# A nascent peptide code for translational control of mRNA stability in human cells

Phillip C. Burke[1,2], Heungwon Park[1] & Arvind Rasi Subramaniam [1,2] ✉

Stability of eukaryotic mRNAs is associated with their codon, amino acid, and GC content. Yet, coding sequence motifs that predictably alter mRNA stability in human cells remain poorly defined. Here, we develop a massively parallel assay to measure mRNA effects of thousands of synthetic and endogenous coding sequence motifs in human cells. We identify several families of simple dipeptide repeats whose translation triggers mRNA destabilization. Rather than individual amino acids, specific combinations of bulky and positively charged amino acids are critical for the destabilizing effects of dipeptide repeats. Remarkably, dipeptide sequences that form extended β strands in silico and in vitro slowdown ribosomes and reduce mRNA levels in vivo. The resulting nascent peptide code underlies the mRNA effects of hundreds of endogenous peptide sequences in the human proteome. Our work suggests an intrinsic role for the ribosome as a selectivity filter against the synthesis of bulky and aggregation-prone peptides.

Protein expression is determined by a balance between the translation rate and stability of mRNAs. In human cells, mRNA stability is often regulated by sequence motifs in the 3′ untranslated region such as microRNA-binding sites and AU-rich elements[1]. Additionally, the protein coding region has been recently recognized as a critical determinant of eukaryotic mRNA stability[2,3]. The role of the coding sequence in mRNA stability is best understood in the budding yeast *S. cerevisiae* where poorly translated codons and nascent peptide motifs with positively charged residues can destabilize mRNAs[4–7]. Poorly translated codons have also been implicated in regulation of mRNA stability in several other organisms[8–11].

Coding sequence features regulating mRNA stability in human cells are less clear. Several recent studies examined the coding sequence determinants of endogenous mRNA stability in human cells and arrived at differing conclusions. Two studies implicated synonymous codon choice as the primary determinant of mRNA stability in human cells[12,13]. Another found GC and GC3 (wobble base GC) content as major factors regulating mRNA stability[14]. A fourth study identified amino acid content to be an important contributor[15]. Extended amino acid motifs and G-quadruplexes in coding regions have also been implicated as triggers of specific mammalian mRNA decay pathways[16,17]. The associations reported in these studies relied on endogenous human coding sequences. Since human mRNAs differ from each other in codon, amino acid, and GC content as well as in their length and the presence of specific sequence motifs, it is challenging to identify the contribution of each factor to mRNA stability. Further, reporters used in the above studies for validation differ extensively in their nucleotide or amino acid content, which complicates their interpretation.

Here, we developed a massively parallel assay to measure the mRNA effects of thousands of coding sequence motifs in human cells. We designed our assay with the initial goal of systematically delineating the individual contribution of mRNA features implicated in previous studies. Instead, we unexpectedly uncovered a potent role for the sequence and structure of the nascent peptide in regulating mRNA stability and ribosome elongation rate. The resulting nascent peptide code regulates the mRNA effects of hundreds of endogenous peptide sequences from the human proteome. Our results point to an unappreciated role for the ribosome as a selectivity filter against the synthesis of bulky and aggregation-prone peptide sequences.

## Results

### A massively parallel assay for mRNA levels in human cells

We reasoned that coding sequence motifs that alter mRNA stability should be identifiable through their effects on steady state mRNA

[1]Basic Sciences Division and Computational Biology Section of the Public Health Sciences Division, Fred Hutchinson Cancer Center, Seattle, WA 98109, USA. [2]Department of Microbiology, University of Washington, Seattle, WA 98195, USA. ✉e-mail: rasi@fredhutch.org

levels. To study the effect of coding sequence motifs on mRNA levels in an unbiased manner, we designed a library of 4096 oligonucleotides made of all possible codon pairs (Fig. 1a). We repeated each codon pair as a tandem 8× repeat with the rationale that their effects will be amplified and readily measurable. We cloned the oligonucleotide library as a pool into a dual fluorescence reporter vector separated by 2 A linkers – a design widely used for studying ribosome stalling motifs in human cells[18–23]. We added multiple random 24nt barcodes without stop codons 3′ of each oligonucleotide insert and linked the barcode sequences to the corresponding insert by high-throughput sequencing. Most studies of coding sequence motifs use transient transfection or lentiviral integration of reporters, which makes measurement of steady state effects on mRNA levels across a large pool difficult. To avoid this, we stably integrated the reporter pool at the *AAVS1* locus of HEK293T cells using CRISPR Cas9-mediated homologous recombination. We extracted mRNA and genomic DNA from the pooled cells and counted each barcode by high-throughput sequencing. Normalization of the total barcode count in the mRNA by the corresponding count in the genomic DNA for each of the 4096 inserts provides a relative

measure of the steady-state mRNA level of that insert. We examined whether our assay captured the effects of known mRNA-destabilizing motifs. We first calculated the effect of individual codons on mRNA level, by averaging across all possible neighboring codons as well as across the first and second positions of each codon within the repeat (Supplementary Fig. 1a). Stop codons in either the first or second position of the codon pair repeat decrease mRNA levels (Fig. 1b, Supplementary Fig. 1a), consistent with their mRNA destabilizing effect due to nonsense-mediated decay[24–26]. We also observe a mild correlation between our measured effects of codons on mRNA level and published codon stabilization coefficients calculated from endogenous mRNA stability (Supplementary Fig. 1c)[15]. However, mRNA levels in our assay show little correlation with GC and GC3 content (Supplementary Fig. 1b) or with binary measures of codon optimality (Supplementary Fig. 1e)[12–15]. Instead, the strongest differences in mRNA abundance in our assay are seen at the amino acid level, with effects spanning a 2-fold range in relative abundance (Fig. 1b). Among the twenty amino acids, the positively charged amino acids lysine and arginine cause the largest average decreases in mRNA levels (Fig. 1b).

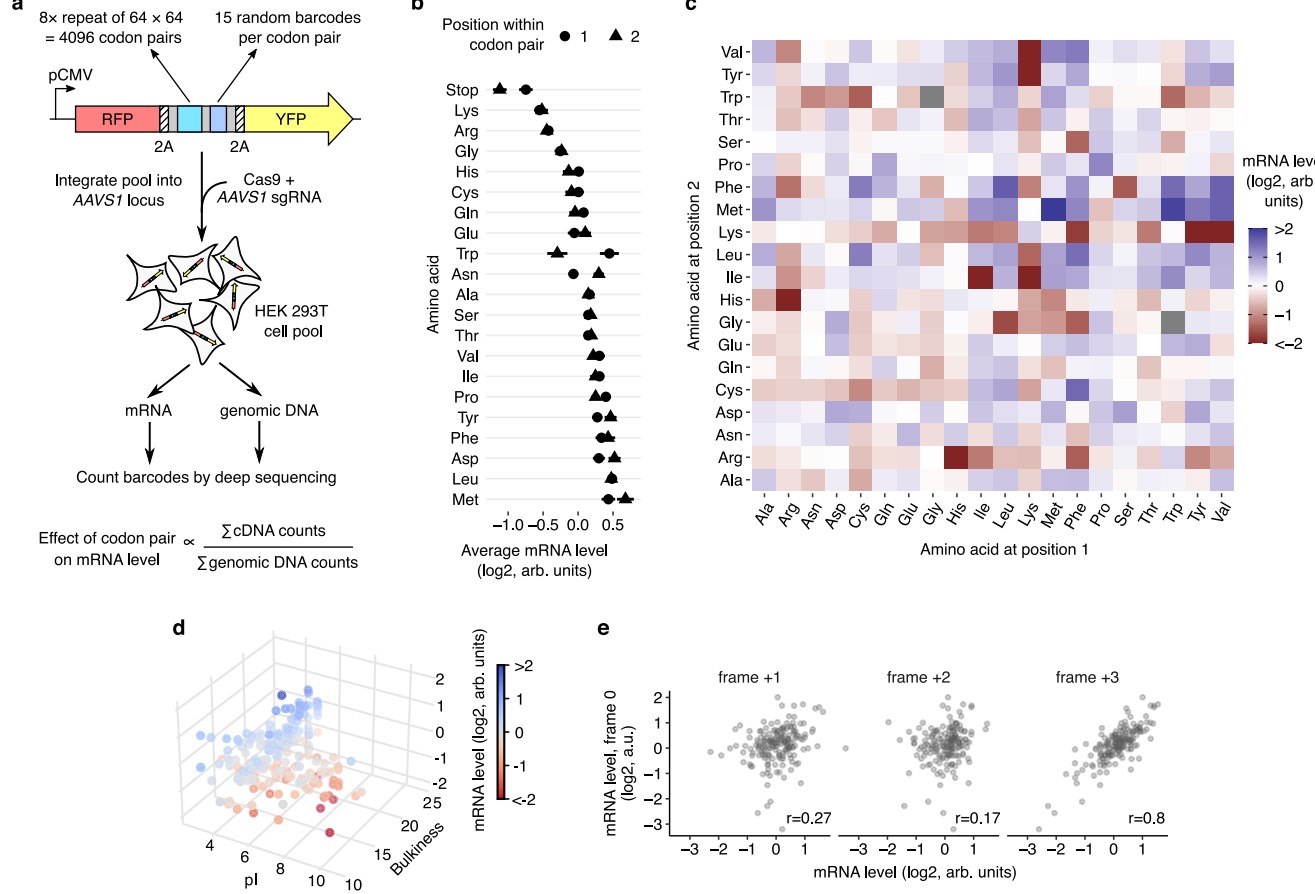

**Fig. 1 | Dipeptide repeats reduce mRNA levels. a** Schematic of massively parallel assay for measuring reporter mRNA levels. 8× repeats of all 4096 codon pairs are synthesized as pooled oligonucleotides, linked in-frame to 24nt random barcodes, and cloned between RFP and YFP reporters with intervening 2 A sequences. Each insert has a median of 15 random barcodes without in-frame stop codons. Reporter cassettes are integrated as a pool at the *AAVS1* locus in HEK293T cells by Cas9-mediated homologous recombination and constitutively expressed off the CMV promoter. Steady state mRNA level of each insert is determined by sequencing corresponding barcodes in the cDNA and the genomic DNA and normalizing the summed cDNA read counts by the genomic DNA read counts. **b** mRNA level of reporters with codons encoding one of the twenty amino acids or a stop codon in position 1 (circles) or position 2 (triangles) of the 8× codon pair insert shown in (**a**). **c** mRNA level of reporters encoding 400 different dipeptide repeats. Amino acids

encoded by the first or second position in the dipeptide are shown along the horizontal or vertical axis respectively. Two dipeptide repeats with missing values are shown in grey. **d** mRNA level of dipeptide repeat-encoding reporters plotted as a function of the average isoelectric point (pI) and the bulkiness[32] of the two amino acids in the dipeptide. **e** mRNA level of reporters encoding 190 different dipeptide repeats (excluding reversed repeats) in the correct reading frame (frame 0, vertical axis) or in reading frames shifted by +1, +2, or +3 nucleotides (horizontal axes). *r* is the Pearson correlation coefficient between frame 0 and the frameshifted mRNA levels. mRNA levels in **b**–**e** are in arbitrary units (arb. units) and are normalized to the median value across all dipeptide repeats. Data in **b** are presented as mean values and error bars represent +/− standard error of measurement (SEM) over a median of 15 barcodes per insert calculated using 100 bootstrap samples. Most error bars in **b** are smaller than data markers.

The known association between positively charged residues in the nascent peptide and slow elongation[27–31] suggests that the decrease in steady-state mRNA levels observed in our assay is caused by ribosome slowdown at these residues.

### Specific dipeptide repeats trigger decrease in mRNA levels

We wondered whether the average effects of amino acids on mRNA levels (Fig. 1b) belie larger effects driven by specific amino acid combinations. We assessed the effect of each pairwise amino acid combination on mRNA abundance and found that these combinations span over a 16-fold range in relative abundance in our assay (Fig. 1c). While lysine and arginine reduce mRNA levels on average, unexpectedly, these amino acids have mild or no negative effect on mRNA levels on their own (Fig. 1c: Lys-Lys, Arg-Arg, Arg-Lys). Rather, the effects of lysine and arginine are primarily driven by co-occurrence with bulky amino acids[32] (ratio of side chain volume to length > 18Å²) such as valine, isoleucine, leucine, phenylalanine, and tyrosine (Fig. 1c). Likewise, most bulky amino acids decrease mRNA levels in combination with lysine and arginine, but not on their own (Fig. 1c). Further, a few dipeptides that contain certain positively charged amino acids (Arg-His) or bulky amino acids (Phe-Ser) also have a strong negative effect on steady-state mRNA levels (Fig. 1c). The combinatorial effect of positively charged and bulky amino acids on mRNA level is captured by a linear statistical model (Fig. 1d): Isoelectric point[32] (pI, a measure of positive charge) and bulkiness[32] of amino acids are positive correlates of mRNA level, while an interaction term between these two physical properties is a negative correlate of mRNA level [mRNA = $(0.31 \times pI)$ + $(0.20 \times bulkiness) - (0.03 \times pI \times bulkiness)$, Adjusted $R^2 = 0.25$]. By contrast, ignoring the interaction between pI and bulkiness results in negative or no correlation of these properties with mRNA level (mRNA = $-0.18 \times pI$, Adjusted $R^2 = 0.21$), which is in line with Fig. 1b. The effects of dipeptide repeats in the translated +0 frame strongly correlates with the codon-matched +3 frame, but only weakly with the codon-mismatched +1 and +2 frames (Fig. 1e). The high correlation between the +0 and +3 frames is also seen from the diagonal symmetry of Fig. 1c and arises from similarity of the encoded peptides (for example $(XY)_8$ and $(YX)_8$ are identical except at their termini). These frame correlations are consistent with the mRNA effects arising at the translational level as opposed to transcriptional or RNA processing differences. Together, our results show that translation of bulky and positively charged amino acids is critical for their negative effect on mRNA level.

### Primary sequence of dipeptide repeats regulates mRNA stability

Several observations suggest that translation of specific dipeptide repeats is a general trigger of mRNA instability in human cells. Multiple human cell lines show lower mRNA levels of the same dipeptide repeats relative to their frameshifted controls (HEK293T, HeLa, HCT116, and K562; Fig. 2a), pointing to the generality of the observed effects. Upon actinomycin D treatment to inhibit transcription, transcripts from reporters with mRNA level-reducing dipeptides decay faster than their frameshifted controls (Fig. 2b). This confirms that the decrease in steady-state mRNA levels caused by dipeptide repeats arises from reduction in mRNA stability.

We wondered if translation of dipeptide inserts that reduce mRNA levels and mRNA stability also cause premature translation termination[18]. To test this, we used fluorescence-activated cell sorting followed by genomic DNA barcode sequencing (FACS-seq) on the 8× codon pair library (Fig. 1a). This reporter library encodes 2A-linked upstream RFP and downstream YFP cassettes surrounding the variable dipeptide sequence, such that inserts that cause premature translation termination will produce RFP but not YFP fluorescence signal (Fig. 1a). We sorted cells that had low YFP signal relative to RFP (low-YFP gate in Fig. 2c and Supplementary Fig. 4d), and then measured the enrichment of each dipeptide insert in this low-YFP population relative to the unsorted population (Fig. 2d). Inserts encoding stop codons between RFP and YFP are enriched in the low-YFP population, indicating that our assay robustly identifies inserts that cause premature termination (Fig. 2d). Similarly, inserts with lower mRNA levels (<2-fold below median in Fig. 1c) are also significantly enriched in the low-YFP gate relative to all other dipeptide inserts (Fig. 2d), indicating that such inserts also cause premature termination in addition to reducing mRNA levels.

Finally, to decipher the effect of dipeptide repetition on mRNA levels, we systematically varied the number of several destabilizing dipeptides identified in our initial assay (Fig. 2e). As the number of dipeptide repeats increases from 1 to 8, each dipeptide starts decreasing reporter mRNA levels at a distinct repeat number between 4 and 7 (Fig. 2e). We then altered the periodicity of dipeptide repeats by intermixing dipeptides with their reversed counterparts such that the overall amino acid composition remains unchanged (Fig. 2f). Even minor perturbations of RH repeats abrogate their negative effect on mRNA levels (Fig. 2f). By comparison, VK repeats had a gradual negative effect on mRNA levels as their periodicity is increased, while SF repeats show an intermediate trend (Fig. 2f). These experiments reveal that the primary sequences of destabilizing dipeptide repeats encode critical regulatory information beyond the identity of the amino acid pairs forming the repeats.

### Secondary structure of dipeptide repeats mediates mRNA effects

Since dipeptide sequences are known to form distinct secondary structures based on their periodicity[33,34], we asked whether mRNA-destabilizing dipeptide repeats adopt specific secondary structures. Using a deep neural network model for secondary structure prediction[35], we find that many dipeptide repeats that strongly reduce mRNA levels in vivo are computationally predicted to form β strands with a high probability (Fig. 3a). We next assigned all dipeptide repeats in the library to either α helices or β strands if their respective prediction probabilities are greater than 0.5. We find that dipeptide repeats predicted to form β strands have a significantly lower mRNA level on average than those predicted to form α helices (Fig. 3b, $P < 0.001$, two-sided Mann-Whitney test). This observation is consistent with the destabilizing amino acids lysine and arginine predominantly occurring in β strands or unstructured peptides in our library (Supplementary Fig. 2a). Among dipeptides containing the positively charged amino acids lysine or arginine, the measured propensity of the second amino acid to occur in a β strand[36] ('Chou-Fasman propensity') is highly correlated with mRNA instability (Fig. 3c; Supplementary Fig. 2b). This correlation is not observed with α helix propensities of the same amino acids (Fig. 3c; Supplementary Fig. 2b), suggesting that β strand formation promotes mRNA instability, as opposed to α helix formation stabilizing mRNAs in our assay. mRNA levels of dipeptide repeats containing the negatively charged amino acid glutamate, which are also predicted to form β strands with high probability when combined with bulky amino acids (Supplementary Fig. 2c), do not show significant correlation with β strand or α helix propensities (Supplementary Fig. 2d). Thus, a combination of bulky and positively charged amino acids in the primary sequence and β strand in the secondary structure are strong and significant predictors of the mRNA-destabilizing effects of dipeptide repeats [mRNA = $(0.30 \times pI) + (0.23 \times bulkiness) - (0.03 \times pI \times bulkiness) - (0.52 \times β$-strand-propensity), Adjusted $R^2 = 0.27$].

### Extended β strands slow ribosome elongation and reduce mRNA levels

To test the causal role of β strands in nascent peptide-mediated translational control, we combined the mRNA-destabilizing dipeptides VK, KV, SF, and FS into 16 amino acid-long peptides. Even though the four constituent dipeptides are strongly predicted to form β strands

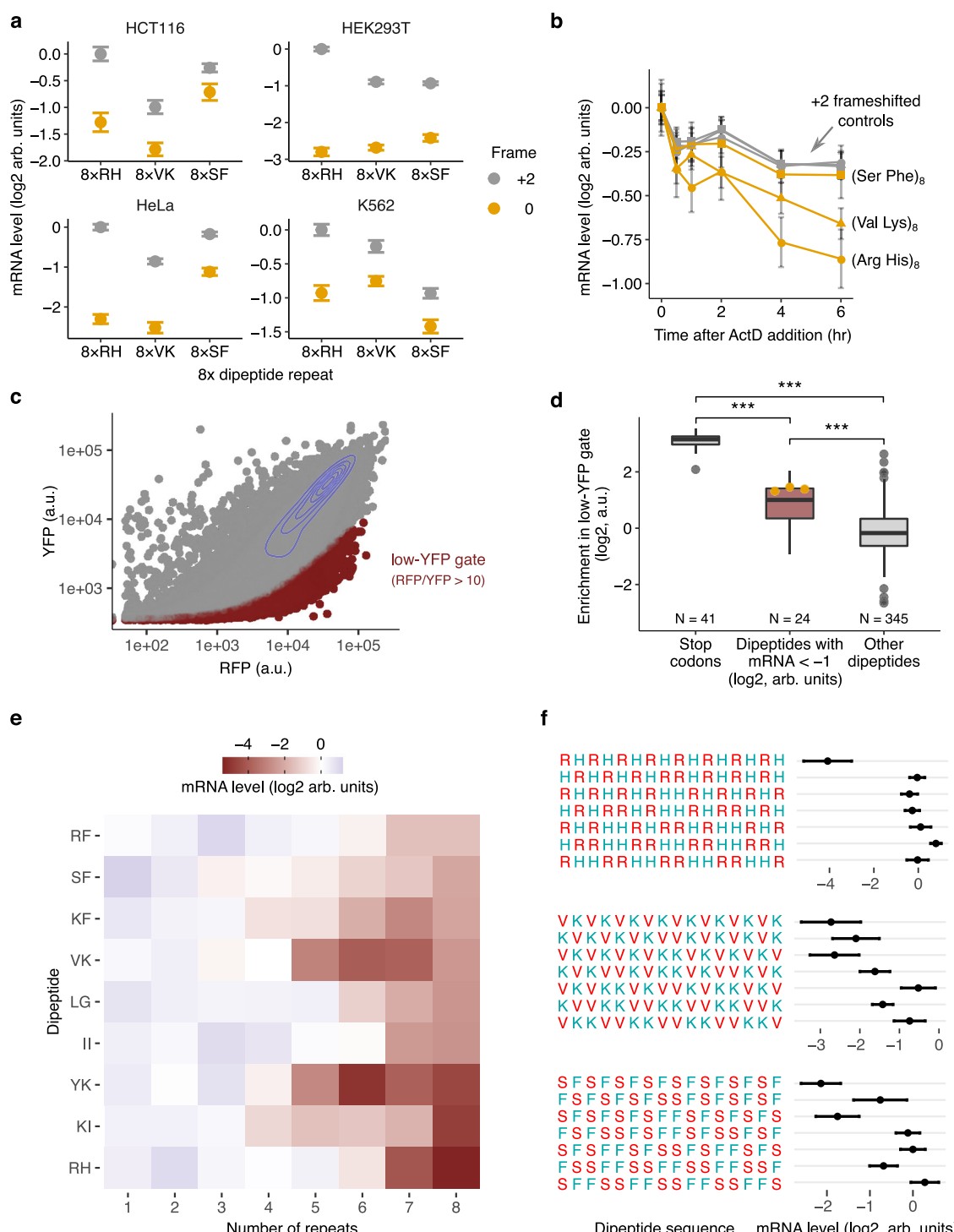

on their own (Fig. 3a), their combinations can form either β strands or α helices with high probability (Fig. 4a). Importantly, all combinations are encoded by the same set of four amino acids to control for amino acid composition. We commercially synthesized two 16 amino acid peptides and used circular dichroism to confirm their secondary structure in vitro (Fig. 4b, left panel). As predicted (Figs. 4a), 4×SVKF primarily forms β strands in aqueous solution, while 4×SKVF forms α helices in the presence of trifluoroethanol (TFE) as a co-solvent[37–39] (Fig. 4b, right panel). We then measured the transit time of ribosomes on mRNAs encoding 16 amino acid inserts preceding a nanoluciferase reporter in a rabbit reticulocyte lysate (RRL) in vitro translation system (Fig. 4c). The β strand-forming 4×SVKF and 4×VKFS inserts slow ribosome elongation relative to the α helix-forming 4×SKVF and 4×KVFS

inserts, with a 200 s difference in in vitro transit time (Fig. 4c). Strikingly, all β strand peptides decrease mRNA levels over 8-fold relative to α helix controls when tested in vivo using our reporter assay (Fig. 4d). We observe similar effects on mRNA level due to β strand formation in HeLa, HCT116, and K562 cells (Supplementary Fig. 3a). We also tested the translation kinetics of the β stranded 8×VK insert by RRL nanoluciferase assay and found that this insert slows ribosome transit time by 100 s relative to its frameshifted control (Supplementary Fig. 3b). Thus, nascent peptides that contain positively charged and bulky amino acids and that are predicted to form β strands trigger ribosome slowdown in human cells. This observation agrees with disome profiling results on endogenous mRNAs, where R-X-K motifs (R – Arg, X – any amino acid, K – lysine) are highly enriched at E, P, and A sites

**Fig. 2 | Nascent peptide primary sequence modulates mRNA stability in human cells. a** mRNA levels of reporters with dipeptide repeats (orange): (Arg His)$_8$, (Val Lys)$_8$, (Ser Phe)$_8$, or the three +2 frameshift controls (grey): (Pro Ser)$_8$, (Gln Ser)$_8$, (Phe Gln)$_8$ in 4 different cell lines: HCT116, HEK293T, HeLa, and K562. **b** mRNA stability of reporters from **a** in HEK293T cells. Reporter mRNA levels are measured at indicated time points after Actinomycin D-induced transcriptional shut off. Most points for the three frameshift controls overlap with each other. **c** Gating strategy for fluorescence-activated cell sorting of HEK293T cells expressing the stably integrated 8× dicodon library from Fig. 1. Cells with a low ratio of YFP/RFP were sorted into the "low-YFP" bin (dark red points). **d** Enrichment of dipeptide inserts in the low-YFP gate is shown for three subgroups; inserts encoding in-frame stop codons, dipeptide repeats with mRNA level < −1 (log2, a.u.) in Fig. 1c, and all other inserts. Enrichment values for the 8×RH, 8×VK, and 8×SF dipeptides are highlighted in orange (8×RH, 8×VK, 8×SF; left to right). The bounds of the box plots are the upper and lower quartile with the median value in the center. Whiskers extend to the most extreme data point no more than [1.5] times the length of the box away from the box. Outliers extending further than the whiskers are shown as individual data points. ***$P$ < 0.001 (two-sided Mann-Whitney U test) for differences between subgroups. Stop vs mRNA < −1 log2 arbitrary units; $P$ value = 2e-16. Stop vs Other; $P$ value = 2e-16. mRNA < −1 log2 arbitrary units vs Other; $P$ value = 2e-7. **e** mRNA levels of dipeptide-encoding reporters with different dipeptide repeat length. Missing values shown in grey. **f** mRNA levels of dipeptide-encoding reporters with different dipeptide repeat periodicity. mRNA levels are measured using the pooled sequencing assay in Fig. 1a and normalized by the median value across all inserts in the pool. Amino acids in **a**, **e**, and **f** are labeled by their one-letter codes. Data in **a** and **b** are mean values ± SEM, calculated over a median of 550 and 370 barcodes per insert respectively, using 1000 bootstrap samples each. Data in **f** are mean values ± SEM, calculated over a median of 15 barcodes per insert using 100 bootstrap samples.

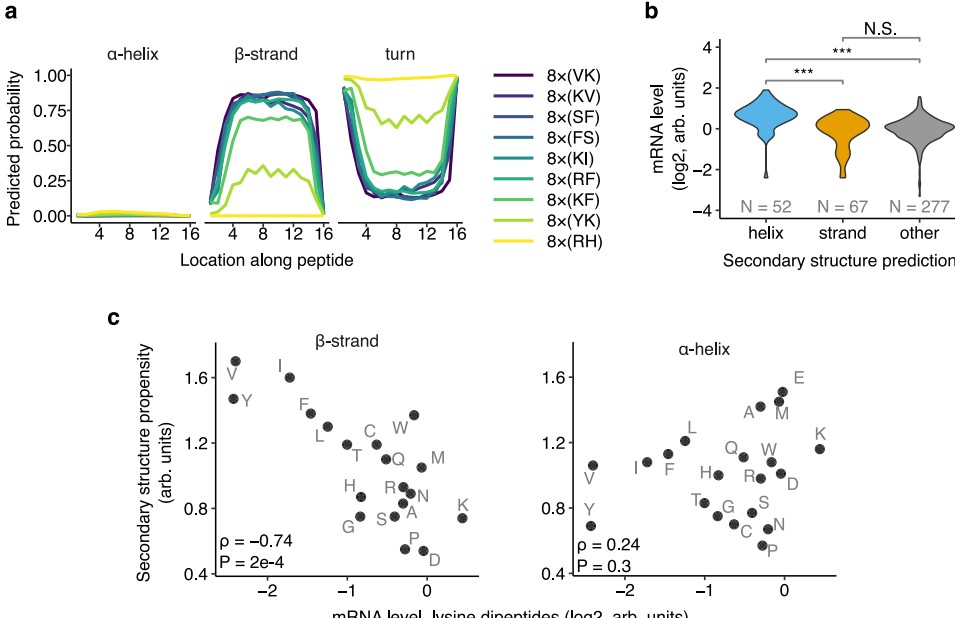

**Fig. 3 | Secondary structure of dipeptide repeats mediates effects on mRNA levels. a** Computationally predicted secondary structure probability along 16 amino acid-long peptide sequences encoded by destabilizing dipeptides. Secondary structure probabilities are predicted using S4PRED[35]. **b** Distribution of mRNA levels of dipeptide repeat-encoding reporters from Fig. 1c partitioned by predicted protein secondary structure. Per residue probability of secondary structure formation are predicted using S4PRED. Inserts with >50% average prediction probability of forming α helix or β strand are classified as such, or else grouped as 'other'. $N$ is the number of dipeptide repeats predicted in each category. ***$P$ < 0.001, N.S.: not significant (two-sided Mann–Whitney U test). Helix vs Strand; $P$ value = 2.2e-9. Helix vs Other; $P$ value = 7.3e-14. Strand vs Other; $P$ value = 0.92. **c** mRNA levels of dipeptide repeat-encoding reporters with lysines in one position and one of twenty amino acids in the other position of the repeat (labeled in grey) shown on horizontal axes. Propensity[36] of the second amino acid to occur in a β strand or an α helix is shown on vertical axes. ρ is the Spearman correlation coefficient between the two axes with the indicated $P$ value (two-sided Spearman rank correlation test).

respectively of the lead ribosome[23]. Notably, several R-X-K motifs with the highest disome density have interspersed bulky residues such as phenylalanine, isoleucine, and leucine[23].

**Dipeptide motifs in the human genome reduce mRNA levels**

We sought to identify endogenous sequences in the human genome that regulate mRNA levels based on the dipeptide code identified above. To do this, we scanned all annotated human protein coding sequences for destabilizing dipeptide combinations of bulky and positively charged amino acids (Fig. 5a). Using a heuristic peptide score (Fig. 5a, top), we identified the 16 amino acid long peptide within each coding sequence that has the maximum density of destabilizing dipeptides. To test whether these endogenous motifs above can reduce mRNA levels, we cloned 1201 such motifs into our reporter and measured their mRNA levels by high throughput sequencing (Fig. 5b). Motifs with high destabilizing dipeptide content result in lower mRNA levels than control motifs ($P$ < 0.01, Fig. 5b, left panel). Among destabilizing motifs, those predicted to form β strands result in lower mRNA levels than the remaining motifs ($P$ < 0.05, Fig. 5b, right panel). To confirm the destabilizing role of the specific dipeptides identified in our study, we disrupted them by moving the bulky and positively charged amino acids to opposite ends without changing the amino acid composition in 1079 endogenous motifs (Fig. 5c, top). As predicted, the resulting mutations increase mRNA levels (median log$_2$ ΔmRNA = 0.38) with 783 mutated motifs having significantly higher mRNA levels ($P$ < 0.05) than their wild-type counterparts (Fig. 5c, bottom). Examination of destabilizing motifs with annotated β strand structures in the Protein Data Bank (PDB) shows that these β strands are part of antiparallel β sheets and are significantly longer than the 5-6 residue length of typical β strands[40] (Fig. 5d). Together, these results show that β-stranded endogenous motifs containing bulky and positively charged dipeptides can reduce mRNA levels.

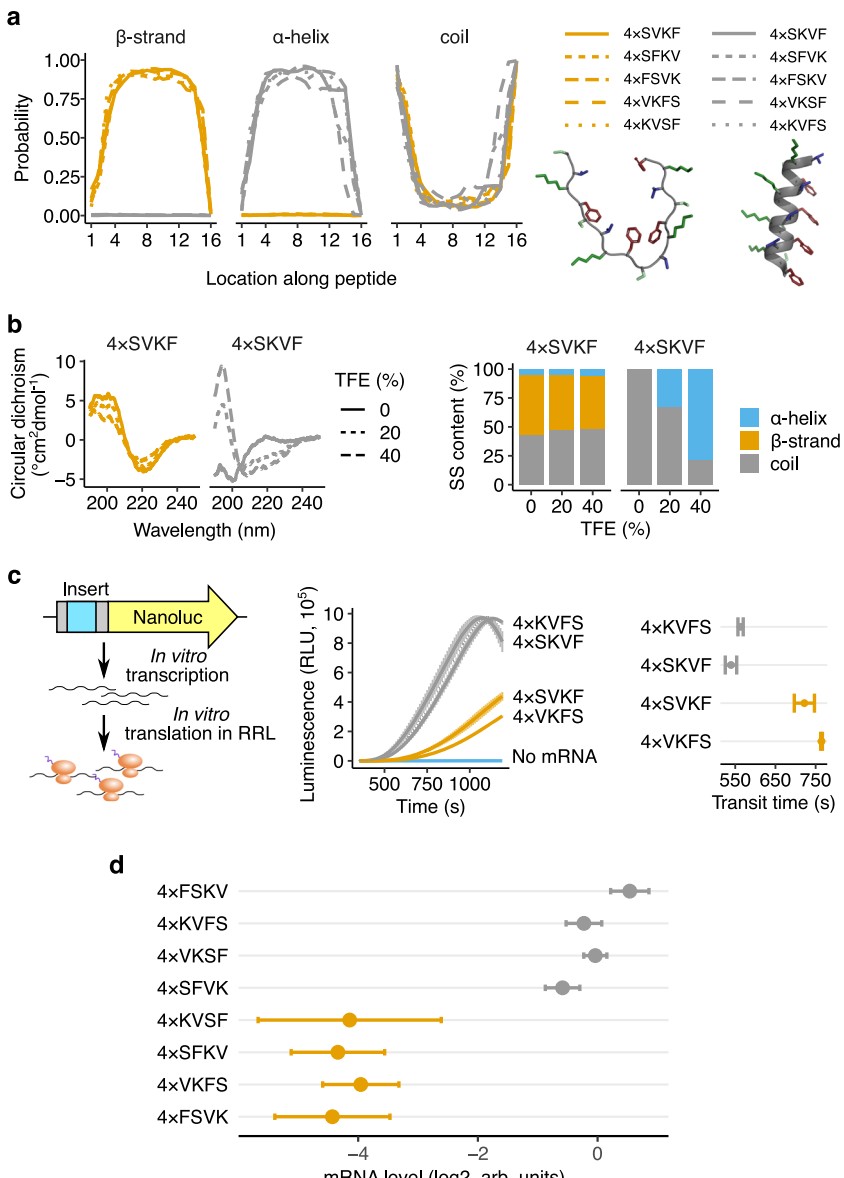

**Fig. 4 | Extended β strands slowdown ribosomes and reduce mRNA levels.**
**a** Computationally predicted secondary structure probability along 16 amino acid-long peptide sequences encoded by alternating VK or KV dipeptides with SF or FS dipeptides identified as destabilizing in Fig. 1. Secondary structure probabilities are predicted using S4PRED. The 10 different peptide sequences are 4× repeats of the dipeptide combinations shown in the legend (for example, 4×SVKF: SVKF SVKF SVKF SVKF). Predicted β strand and α helix structures of 4×SVKF and 4×SKVF respectively using PEP-FOLD3[104] are shown below the legends. The peptide backbones are in grey and the side chains of amino acids are colored. **b** Measured circular dichroism spectra of in vitro synthesized 4×SVKF or 4×SKVF peptides (left). Measurements are performed with 0, 20, 40% of Trifluoroethanol (TFE) as co-solvent in 10 mM sodium phosphate buffer (pH = 7.5). Relative content of different secondary structures is estimated by linear deconvolution of the measured spectra from a pre-computed basis set using SESCA[105]. **c** In vitro measurements of ribosome transit time on mRNAs encoding β strand- or α helix-forming peptides followed by Nanoluciferase. Luminescence is measured as a function of time after addition of in vitro transcribed mRNAs to rabbit reticulocyte lysate (RRL) at t = 0 s. Standard error of measurement across three technical replicates is shown as a shaded area on either side of the mean. Ribosome transit times are estimated by measuring the X-intercept of the linear portion of the raw luminescence signal. **d** In vivo mRNA levels of reporters encoding one of eight different dipeptide combinations. mRNA levels are measured using the reporter constructs and pooled sequencing assay in Fig. 1a. Data are presented as mean values ± SEM over a median of 550 barcodes per insert calculated using 1000 bootstrap samples.

## Discussion

Here, we identify a combinatorial code composed of bulky, positively charged, and extended β strand nascent peptides that regulates translation and mRNA stability in human cells. We demonstrate that a minimal combination of these sequence and structural elements is sufficient to induce ribosome slowdown and cause changes in gene expression, and is widespread in the human proteome. As discussed below, elements of the code uncovered here allow us to synthesize a large body of observations on nascent peptide-mediated slowdown of ribosomes and regulation of mRNA stability in human cells. Our results also point to a role for the ribosome as a post-synthesis filter against nascent peptide sequences that are bulky and aggregation prone.

The nascent peptide code for mRNA stability described here is significantly more complex and localized along the mRNA than previously associated sequence features such as codons, amino acids, and GC content[12–15]. We don't observe large effects on mRNA levels due to codon optimality or GC content in our assay (Supplementary Fig. 1). This is likely because the 48 nucleotide inserts constitute only ~3% of the 1725 nucleotide coding sequence of our library reporters (Fig. 1a), which limits the impact changing these motifs can have on overall

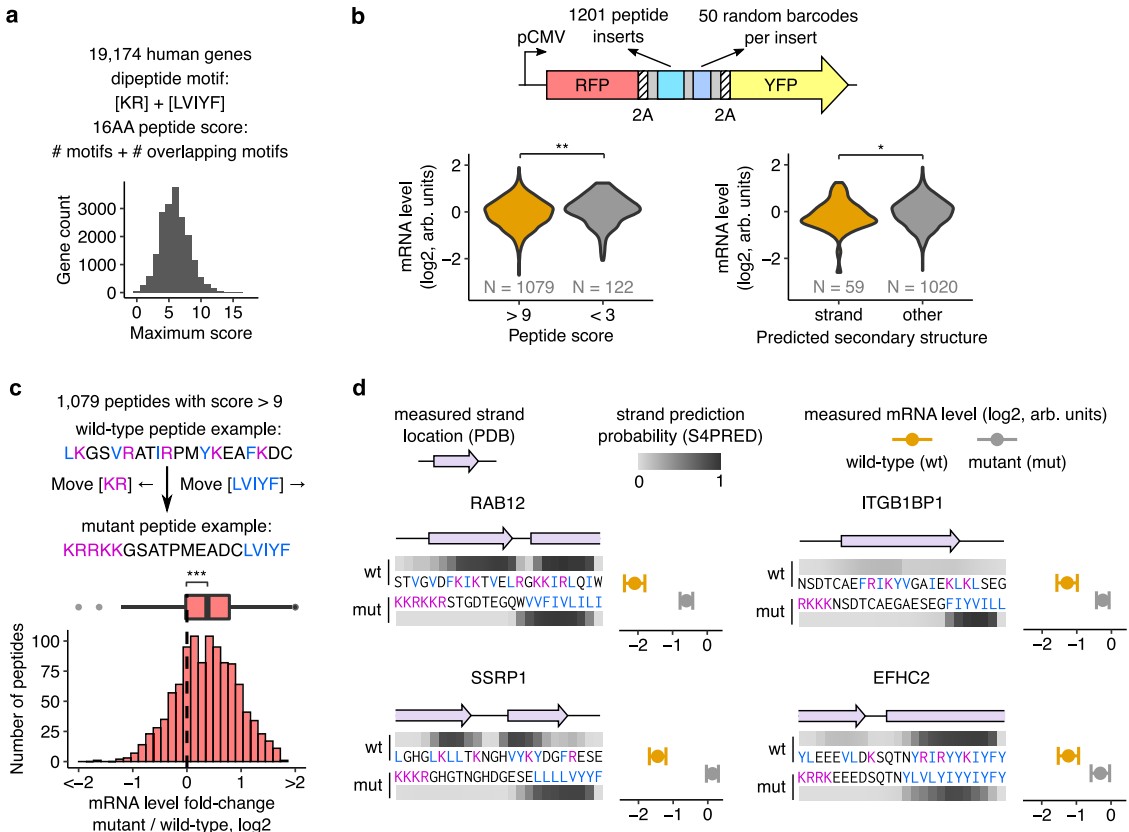

**Fig. 5 | Dipeptide motifs in the human genome reduce mRNA levels. a** Scoring methodology for dipeptide motifs in human CDS. Destabilizing dipeptides formed by lysine (K) or arginine (R) with an adjacent leucine (L), valine (V), isoleucine (I), tyrosine (Y), or phenylalanine (F) are given a score of 1. If two such dipeptides overlap, an additional score of 1 is given. The 16 amino acid peptide window with the maximum score is identified in each CDS, and the distribution of these peptide scores across all genes is shown in the lower panel. **b** mRNA levels of destabilizing (peptide score >9) and control motifs (peptide score <3) are measured by pooled cloning (1201 total inserts) into a reporter construct followed by deep sequencing as in Fig. 1a. Left panel: Distribution of measured mRNA levels of destabilizing dipeptide motifs compared to control motifs (P = 0.004). Right panel: Distribution of measured mRNA levels of destabilizing dipeptide motifs in the left panel partitioned by predicted secondary structure (P = 0.022). 59 motifs with an average β strand prediction probability >0.5 using S4PRED are classified as β strands.

**c** Increase in measured mRNA levels upon reordering amino acids in 1,079 endogenous destabilizing dipeptide motifs from C (median Δlog2 mRNA = 0.38, P = 2.2e-16). All codons encoding K or R are moved to the 5′ end of the mutated motif and codons encoding L, V, I, Y, or F are moved to the 3′ end. mRNA levels of motifs are measured using the pooled reporter assay in **b**. ***P < 0.001, **P < 0.01, *P < 0.05 using two-sided Mann–Whitney U test in **b** and **c**. **d** Examples of endogenous destabilizing motifs with known β-stranded secondary structure. The measured secondary structure of each wild-type motif from PDB is shown as a purple ribbon diagram. Prediction probability for β strands using S4PRED is shown as a grayscale heatmap for wild-type and mutant motifs. Measured mRNA levels of wild-type (orange) and mutant (grey) motifs are shown on the right within each panel. mRNA levels of wild-type and mutant motifs are measured using the pooled reporter assay in **c**, and presented as mean values ± SEM over a median of 50 barcodes per insert calculated using 1000 bootstrap samples.

reporter composition. Nevertheless, some individual codon and amino acid signatures in our data agree with the findings of previous studies (Fig. 1b, Supplementary Fig. 1). For example, bulky amino acids such as Leu, Ile, Val, and Phe are stabilizing on average, though their codon-specific effects vary across previous studies[12,13,15]. The amino acid serine shows prominent codon-specific effects, with AGU and AGC codons reducing mRNA level more than the remaining codons[12–15]. The methionine AUG start codon and the near-cognate start codons (CUG, GUG, UUG) all promote mRNA stability[12–15], possibly through effects on increased downstream translation[41]. With the exception of arginine, lysine, and glycine, our amino acid level effects correlate with the amino acid stability coefficient calculated from endogenous mRNA stability (Supplementary Fig. 1d)[15]. While glycine codons generally stabilize endogenous mRNAs in prior studies, all four glycine codons decrease mRNA levels in our assay, suggesting that glycine dipeptides also cause nascent peptide-mediated ribosome slowdown and mRNA instability. Indeed, we find that Gly-Gly dipeptides reduce mRNA levels (Fig. 1c) consistent with previous observations that poly-glycine motifs slowdown ribosomes[42]. In our data, glycine has the largest effects on mRNA levels when in combination with Leu and Phe, suggesting a

nascent peptide-mediated destabilization mechanism akin to that of the biochemically similar Ser-Phe dipeptides.

While positive charge in the nascent peptide can slow ribosomes[27,28], our results show that positive charge by itself is insufficient to induce changes in gene expression in human cells. The importance of bulky amino acids for mRNA effects observed here is in line with the role of side chain bulk in ribosome-associated quality control in *S. cerevisiae*[43]. Further, bulky synthetic amino acid analogs in the nascent peptide and small molecules that add bulk to the exit tunnel can both reduce ribosome elongation rate[44–47]. Ribosome profiling in *S. cerevisiae* and human cells shows that tripeptide combinations of bulky and positively charged amino acids are enriched at sites of increased ribosome density[23,48]. Bulky and positively charged amino acids also play critical roles in many known ribosome-arresting peptides[7,49–52], and several human arrest peptide sequences stall ribosomes specifically in the presence of small molecule metabolites or drugs in the ribosome exit tunnel[53,54]. Structural studies of arrest peptides suggest that bulky and positively charged amino acids might slow down ribosomes by altering the geometry of the peptidyl-transferase center (PTC) and/or by steric interactions with the

constriction point in the exit tunnel formed by the uL4 and uL22 proteins[7,51,55,56].

Our work shows that extended β strand motifs in nascent peptides contribute to ribosome slowdown and mRNA instability in human cells. This role of a simple secondary structural motif like β strand is surprising given that cryo-EM studies of stalled ribosome nascent chain complexes reveal a diverse range of extended conformations, turns, and helices that are specific to each arrest peptide[7,47,52,57,58]. This comparison is complicated by the fact that cryo-EM studies are performed on post-arrest complexes where the nascent chain might have already undergone extensive conformational rearrangements. Further, while several motifs uncovered here form β strands in silico and in vitro in isolation, they might have a significantly different structure within the confined geometry of the ribosome exit tunnel[39,59–61]. At the molecular level, β strands in nascent chains could contribute to ribosome slowdown as an allosteric relay that communicates steric interactions between the nascent chain and the distal portions of the ribosome exit tunnel such as the uL4/uL22 constriction to the PTC[7,45,56,57,62]. This possibility is supported by our observation that destabilizing dipeptide repeats are at least 10-12 amino acids long (Fig. 2e), which is consistent with the distance between the uL4/uL22 constriction and the PTC.

In addition to the sequence and structural determinants of nascent peptide-mediated ribosome slowdown studied here, several classes of nascent peptide sequences that slowdown ribosomes might not be revealed by our assay. For example, poly-prolines do not emerge as destabilizing motifs in our assay even though they are known to slowdown ribosomes[23]. This is likely because poly-proline stalls are resolved without triggering quality control or mRNA instability[17]. While extended β strands are the primary structural motif associated with ribosome slowdown here, we also find motifs with unstructured regions that nevertheless reduce mRNA levels (Figs. 3b and 5b). This might be in part due to limitations of existing computational methods[35] to predict secondary structures or their limited relevance to secondary structures forming inside the ribosome. It is also likely that the combinatorial code of positively charged, bulky, and β strand sequences uncovered here underlies some, but not all, classes of nascent peptides that have the potential to slowdown ribosomes and effect changes in gene expression. For example, the arginine-histidine dipeptide repeat destabilizes mRNA and causes premature termination similar to the β stranded Val-Lys and Ser-Phe inserts (Fig. 2a–d). Unlike the latter inserts, Arg-His effects require a longer insert length and strict dipeptide periodicity (Fig. 2e, f), and occur with no predicted β strand formation (Fig. 3a). The 8×Arg-His repeats are reminiscent of dipeptide repeat expansions in the human C9ORF72 gene, which cause neurological disease in humans[63–65]. Alternate initiation in the C9ORF72 ORF results in translation of extended Arg-Gly and Arg-Pro repeats, which stall ribosomes and cause premature termination in a length dependent manner, with 20× dipeptide repeats being the minimal length required to to stall ribosomes[66,67]. Unsurprisingly, we do not observe marked effects from 8× Arg-Gly or Arg-Pro in our assay, as 10× repeats of these dipeptides do not cause premature termination[66]. However, to our knowledge, Arg-His dipeptide repeats have never been tested in this manner prior to our work. It may be that Arg-His repeats impact translation through a similar mechanism as Arg-Gly and Arg-Pro repeats, but with a more acute effect on ribosome elongation that requires fewer repeats to trigger. Notably, Arg-rich peptides without Gly or Pro dipeptide periodicity (for example 12×Arg) do not stall ribosomes[66,68]. This agrees with our finding that positively charged dipeptide repeats composed of 8×RR and 8×RK have little effect on mRNA levels (Fig. 1c).

Nascent peptides that slowdown ribosomes might exert their effects on mRNA stability through distinct cellular pathways compared to the ones sensing codon, amino acid, and GC content of mRNAs[22,69–71]. In this vein, poly-lysine sequences encoded by poly-A

and the *Xbp1* arrest sequence are among the few known nascent peptide motifs with intrinsic ability to stall ribosomes in human cells[72–74]. Both poly-A runs and the *Xbp1* arrest sequence are substrates of the ribosome-associated quality control (RQC) pathway, which causes premature translation termination in response to ribosome collisions, limiting production of the proteins encoding these motifs[18–20,23,51]. The RQC pathway is most well studied in yeast, where it also destabilizes the mRNA encoding the stalling motif through activity of the endonuclease Cue2, in a process termed No-Go decay[4,69,75]. While the effects of the human RQC pathway on mRNA stability are not fully characterized, humans have a Cue2 homolog, N4BP2, which suggests that this pathway could reduce mRNA level in addition to limiting protein production[16,76]. There are also examples of pathological peptide repeat sequences that cause ribosome slowdown and premature termination which are not subject to the RQC pathway[66,69]. This includes Arg-Gly and Arg-Pro dipeptides from the C9ORF72 ORF, which cause amyotrophic lateral sclerosis and frontotemporal dementia[63,64], and poly-glutamine repeats translated from CAG nucleotide repeat expansions in the *mHtt* gene, which cause Huntington's disease[77–79]. Interestingly, although the RQC pathway isn't demonstrated to act directly on these toxic repeats, expression of RQC pathway components is associated with lower disease severity in both instances[79,80]. As the destabilizing peptide sequences we identify in this study cause ribosome slowdown (Fig. 4c) and premature termination (Fig. 2d), we suspect that some inserts may be directly repressed by RQC, in a manner similar to the stalling XBP1u nascent protein[51], whereas others may be resistant to RQC repression, as is the case with Arg-rich dipeptides[66]. In addition, it is likely that the effects of endogenous nascent peptide motifs on ribosome slowdown and mRNA stability are modulated by other co-translational events such as nascent protein folding outside the ribosome[81,82], membrane insertion[83,84], and multiprotein assembly[85,86].

The nature of the nascent peptide code uncovered here has important implications for cellular homeostasis and disease. Ribosome slowdown and mRNA destabilization induced by bulky and extended β strands, which are highly aggregation prone[87,88], implies that the ribosome has an intrinsic ability to throttle the synthesis of such proteins. Ribosome slowdown at extended β strands could serve as a quality control mechanism, testing the ability of long β strands (10 amino acids or greater in length) to eventually fold into antiparallel β sheets outside the ribosome, and thus avoid aggregation. This ribosomal selectivity filter would act before other co-translational mechanisms such as codon optimality that help avoid aggregation after β strands emerge from the ribosome[89,90]. Slow translation elongation without mRNA decay can also help recruit protein chaperones, which may be important to properly fold β strands[91,92]. Finally, the gene regulatory potential of the dipeptide motifs uncovered here suggests that disease-causing missense mutations occuring at these motifs might exert their phenotype by altering protein expression *in cis* rather than protein activity.

## Methods
### Plasmid construction
Plasmids, oligonucleotides, and cell lines used in this study are listed in Supplementary Data 1.

### Parent vector construction
The AAVS1-targeting parent vector pPBHS285 used for this study was constructed using Addgene plasmid #68375[93] as a backbone. The PGK1 promoter was replaced with the CMV promoter and the native pCMV 5' UTR region. The coding sequence was replaced by a codon-optimized mKate2 and eYFP fusion cassette, linked with two 2A linker sequences. These 2A sequences surround a cassette encoding an EcoRV restriction site, Illumina R1 sequencing primer binding site, and a T7 promoter. The R1 primer binding and T7 sequences are for sequencing of inserts

and barcode sequences cloned at the EcoRV site and for in vitro transcription from genomic DNA, respectively.

## Variable oligo pool design
Four oligo pools were designed for this study.

Pool 1 (Fig. 1b–e, Fig. 3a,c) encodes all possible dicodon (6nt) combinations, for a total of 4096 codon pairs. These 6nt dicodon inserts were repeated eight times to create 8× dicodon repeat inserts, each 48nt in length.

Pool 2 (Figs. 2e, f and 4d) encodes several dipeptide combinations identified in Library 1 as reducing mRNA levels. For Fig. 2e, the number of dipeptide repeats was systematically reduced from 8 to 1. Repeats were replaced with a Ser-Gly linker, shown to be not destabilizing in Library 1, to maintain a constant 48nt insert length. For Fig. 2f, periodicity of dipeptides was altered by interspersing 1, 2, or 4 tandem repeats of each dipeptide with an equal number of its sequence-reversed counterpart. For Fig. 4d, destabilizing dipeptides KV and SF were combined and rearranged to form either α helices or β strands, as predicted by S4PRED[35].

Pool 3 (Fig. 5) encodes 16 amino acid nascent peptide motifs from the human proteome identified as potentially destabilizing by the scoring method described in Fig. 5a along with 4 flanking codons on either side. The library encodes the top 1079 predicted stalling motifs with a peptide score >9, and 122 control motifs with a peptide score <3. The library also includes the mutants with reordered amino acids from the 1079 endogenous destabilizing dipeptide motifs, which were designed as shown in Fig. 5c.

Pool 4 (Fig. 2a, b, Supplementary Fig. 3a) encodes 8 inserts: 3 destabilizing dipeptide repeats $(RH)_8$, $(VK)_8$, $(SF)_8$, their respective frameshift controls $(PS)_8$,$(QS)_8$,$(FQ)_8$, the β strand peptide $(SVKF)_4$, and the α helix peptide $(SKVF)_4$.

Oligo pools 1–3 were synthesized by Twist Biosciences with flanking sequences for PCR and cloning into the EcoRV site of the parent pPBHS285 vector. Oligo pool 4 was cloned by PCRing individual inserts and pooling them before cloning.

## Plasmid library construction
Parent vector pPBHS285 was digested with EcoRV. The oligo pools described above were PCR amplified using primers oHJ01 and either oPB348 (Library 1) or oPB409 (Libraries 2–4). oPB348 and oPB409 both encode a 24 nt random barcode region, comprised of 8×VNN repeats to exclude in-frame stop codons (where V is any nucleotide except T). Barcoded oligo pools were cloned into pPBHS285 by Gibson assembly. Assembled plasmid pools were transformed with high efficiency into NEB10Beta *E.coli*. For pools 1–3, the transformed plasmid pools were extracted from 15-50 *E.coli* colonies per insert in the library, thus bottlenecking the number of unique barcodes present in each plasmid pool. Resulting plasmid pools contained between 60,000–400,000 unique barcode sequences for pools 1–3. For pool 4, the transformed library was bottlenecked to around 150 barcodes per insert, and 6 such pools with distinct barcodes were extracted for multiplexed library preparation of different cell lines. The plasmid libraries corresponding to pools 1-4 are pPBHS286, pPBHS309, pHPHS296, and pHPHS406, respectively. Variable insert and barcode sequences for each plasmid library are provided as part of the data analysis code.

## CRISPR vectors
The CLYBL-targeted Cas9-BFP expression vector pHPHS15 was constructed by Golden Gate assembly of either entry plasmids or PCR products with pHPHS11 (MTK0_047[94] Addgene #123977) as backbone, pHPHS3 (MTK2_007[94] Addgene #123702) for the pEF1a promoter, pADHS5[95] (pU6-(BbsI)_CBh-Cas9-T2A-BFP[96] Addgene #64323) for the Cas9-2A-BFP insert cassette, and pHPHS6 (MTK4b_003[94] Addgene #123842) for the rabbit β-globin terminator. sgRNA vectors pPBHS320 (gRNA_AAVS1-T1 Addgene #41817) and pADHS4[95] (eSpCas9(1.1) _No_FLAG_AAVS1_T2 Addgene #79888) were used for insertion at the AAVS1 locus. pASHS16 (MTK234_030 spCas9-sgRNA1-hCLYBL[94] Addgene #123910) was used for insertion at the CLYBL locus.

## Cell line maintenance and generation
HEK293T cells (RRID:CVCL_0063, ATCC CRL-3216), HCT116 cells (RRID:CVCL_0291, NCI60 cancer line panel), and HeLa cells (RRID:CVCL_0030, ATCC CCL-2) were grown in DMEM (Thermo 11965084). K562 cells (RRID:CVCL_0004, ATCC CCL-243) were grown in IMDM (Thermo 12440053). Media for all cells was supplemented with 10% FBS (Thermo 26140079). Cells were grown at 37 C in 5% $CO_2$. All transfections into HEK293T, HCT116, and HeLa cells were performed using Lipofectamine 3000 (Thermo L3000015). Transfections into K562 cells were performed using an Amaxa Nucleofector V kit (Lonza VCA-1003). HEK293T cells that stably express Cas9 (hsPB80) were generated by transfecting the CLYBL::Cas9-BFP vector pHPHS15 and spCas9 sgRNA1 hCLYBL vector, and selecting with 200 μg/mL hygromycin.

## CRISPR integration of plasmid libraries
hsPB80 CLYBL::Cas9-BFP HEK293T cells were seeded to 50% confluency on 15 cm dishes for all library transfections. 10 μg of library plasmid (pPBHS286, pPHBS309, or pHPHS296) and 1.5 μg of each AAVS1 targeting CRISPR vector were transfected per 15 cm dish. pPBHS286, and pPBHS309 were each transfected into a single 15 cm dish. pHPHS296 was transfected into three 15 cm dishes. pHPHS406 pools with different barcodes were transfected into single 10 cm dishes of hsPB80, HeLa, HCT116 and 2 million cells of K562. Cells were selected with 2 μg/mL puromycin, added 48 h post-transfection. Cells from the three pHPHS296 transfections were combined at the start of selection. Puromycin selection was removed after 6–10 days, once cells were growing robustly in selection. 24 h after removing puromycin selection, stable library cells were plated into two separate 15 cm dishes, to reach 75% confluency the next day, for matched mRNA and gDNA harvests. For pHPHS406, libraries were maintained in two 10 cm dishes or T75 flasks (for K562).

## mRNA stability measurement
hsPB80 cells containing the stably integrated pHPHS406 library were seeded to 50% confluence in a 6-well plate. Actinomycin D (ActD) powder was dissolved in DMSO at 1 mM (1.25 mg/mL) and added to each well of the 6-well plate to a final concentration of 5 μg/mL. Before harvesting, 1 million HeLa cells containing the pHPHS406 library were lysed in 6 mL of Trizol reagent, to create a Trizol lysis solution containing a set number of mRNAs with different barcodes than those in the hsPB80 pHPHS406 pool, for barcode count normalization across samples. ActD treated hsPB80 wells were harvested at 0, 0.5, 1, 2, 4, and 6 h after the addition of ActD by adding 0.75 mL of the Trizol lysis solution above to wells at each timepoint, then following the manufacturer's mRNA extraction protocol.

## Library Genomic DNA extraction
Reporter library genomic DNA was harvested from one 75% confluent 15 cm or 10 cm dish of stably expressing library cells. Genomic DNA was harvested using Quick-DNA kit (Zymo D3024), following the manufacturer's instructions, with 3 mL of genomic DNA lysis buffer per 15 cm plate, and 1 ml of the same buffer per 10 cm plate. Between 0.5–10 μg of purified genomic DNA from each library sample was sheared into ~350 nucleotide length fragments by sonication for 10 min on ice using a Diagenode Bioruptor. Sheared gDNA was then in vitro transcribed into RNA (denoted gRNA below and in analysis code) starting from the T7 promoter region in the insert cassette, similar to previous approaches[97,98], using a HiScribe T7 High Yield RNA Synthesis Kit (NEB E2040S). Transcribed gRNA was treated with DNase

I (NEB M0303S) and cleaned using an RNA Clean and Concentrator kit (Zymo R1013).

## Library mRNA extraction

Reporter library mRNA was harvested from one 75% confluent 15 cm or 10 cm dish of stably expressing library cells. mRNA was harvested by using 3 mL of Trizol reagent (Thermo) to lyse cells directly on the plate, and then following the manufacturer's mRNA extraction protocol. Purified mRNA was treated with DNaseI (NEB M0303S) and then cleaned using an RNA Clean and Concentrator kit (Zymo R1013).

## mRNA and genomic DNA barcode sequencing

Between 0.5–10 μg of DNaseI-treated mRNA and gRNA for each library was reverse transcribed into cDNA using Maxima H Minus Reverse Transcriptase (Thermo EP0752) and a primer annealing to the Illumina R1 primer binding site (oPB354). A 170-nucleotide region surrounding the 24-nucleotide barcode was PCR amplified from the resulting cDNA in two rounds, using Phusion Flash High-Fidelity PCR Master Mix mastermix (Thermo F548L). Round 1 PCR was carried out for 10 cycles, with cDNA template comprising 1/10th of the PCR reaction volume, using primers oPB361 and oPB354. Round 1 PCRs were cleaned using a 2× volume of Agencourt Ampure XP beads (Beckman Coulter A63880) to remove primers. Cleaned samples were then used as template for Round 2 PCR, carried out for 5-15 cycles, using a common reverse primer (oAS111) and indexed forward primers for pooled high-throughput sequencing of different samples (oAS112-135 and oHP281-290). Amplified samples were run on a 1.5% agarose gel and fragments of the correct size were purified using ADB Agarose Dissolving Buffer (Zymo D4001-1-100) and UPrep Micro Spin Columns (Genesee Scientific 88–343). Concentrations of gel-purified samples were measured using a Qubit dsDNA HS Assay Kit (Q32851) with a Qubit 4 Fluorometer. Samples were sequenced using an Illumina HiSeq 2500 or Illumina NextSeq 2000 in 1×50, 2×50, or 1×100 mode (depending on other samples pooled with the sequencing library).

## Insert-barcode linkage sequencing

Plasmid library pools 1-4 (pPBHS286, pPBHS309, pHPHS296, and pHPHS406) were diluted to 10 ng/μL. A 240-nucleotide region surrounding the 48-nucleotide variable insert sequence and the 24-nucleotide barcode was PCR amplified from these pools in two rounds, using Phusion Flash High-Fidelity PCR Master Mix mastermix (Thermo F548L). Round 1 PCR was carried out for 10 cycles, with 10 ng/μL plasmid pool template comprising 1/10th of the PCR reaction volume, using primers oPB29 and oPB354. Round 1 PCRs were digested with DpnI (Thermo FD1704) at 37 °C for 30 minutes to remove template plasmid and cleaned using a 2× volume of Agencourt Ampure XP beads (Beckman Coulter A63880) to remove primers and enzyme. Cleaned samples were used as template for Round 2 PCR, for 5 cycles, using oAS111 and indexed forward primers (oAS112-135 and oHP281-290). Amplified Round 2 PCR products were purified after size selection and quantified as described above for barcode sequencing. Samples were sequenced using an Illumina MiSeq or Illumina NextSeq 2000 in 2 × 50 or 1 × 100 mode.

## Fluorescence-activated cell sorting and genomic DNA sequencing assay

Two 15 cm dishes of 75% confluent hsPB80 cells stably expressing the pHPHS286 library were used as input for fluorescence-activated cell sorting, using a BD FACSAria II flow cytometer. Fluorescence values of the first 50,000 sorted cells are plotted for reference in Fig. 2c. Fluorescence gates were determined using hsPB80 cells containing the pHPHS285 no-insert parent vector and untransfected hsPB80 cells as positive and negative controls for RFP and YFP fluorescence. Full gating strategy for the pHPHS286 library cells and pHPHS285 no-insert cells is in Supplementary Fig. 4). 2.5 M cells with -10-fold or greater RFP

expression relative to YFP were sorted into the low-YFP gate and gDNA was extracted from these cells, as well as from 2.5 M unsorted cells from the same suspension, using 3 mL of gDNA lysis buffer. 4 μg of gDNA from each sample was used as input for gDNA barcode sequencing, following the procedures detailed above. Barcodes in each sample were quantified as described in the computational methods below. Low-YFP gate enrichment for each dipeptide insert was calculated as the log2 ratio of the summed low-YFP barcode counts to the summed unsorted barcode counts.

## Rabbit reticulocyte nanoluciferase transit time assay

DNA fragments encoding 4×KVFS and 4×SKVF (α helix) and 4×VKFS and 4×SVKF (β strand) peptides were generated by PCR-amplifying overlapping oligos that encode each sequence in the forward and reverse direction (oPB470-473 and oPB488-491). Nanoluciferase cassette was amplified from an IDT gBlock (oPN204) using oAS1287 and oPB465. Insert sequences and the Nanoluciferase cassette were combined by overlap PCR using oPB464 and oPB462, which add a 5′ T7 promoter site and a 3′ polyA tail to the amplified reporter template, with oAS1545 used to bridge oPB462 annealing. Resulting insert-Nanoluciferase cassette sequences were confirmed by Sanger sequencing. The PCR products were transcribed into mRNA using a HiScribe T7 High Yield RNA Synthesis Kit (NEB E2040S). mRNA was cleaned using an RNA Clean and Concentrator kit (Zymo R1013). In vitro Nanoluciferase reporter translation reactions were performed as described in Susorov et al. 2020[99]. Reaction mixture containing 50% of nuclease-treated rabbit reticulocyte lysate (RRL) (PRL4960, Promega) was supplemented with 30 mM Hepes-KOH (pH = 7.5), 50 mM KOAc, 1.0 mM Mg(OAc)$_2$, 0.2 mM ATP and GTP, 0.04 mM of 20 amino acids (PRL4960, Promega), and 2 mM DTT. Nanoluciferase substrate furimazine (PRN1620, Promega) was added to the mixture at 1%. 15 μL aliquots of the mixture were placed in a 384-well plate and incubated at 30 °C for 5 min in a microplate reader (Tecan INFINITE M1000 PRO). Translation reactions were started by simultaneous addition of 3 μL mRNA, to a final concentration of 10 ng/μL, and luminescence signal was recorded every 10 seconds over a period of 25 minutes.

## Circular dichroism

4×SKVF (α helix) and 4×SVKF (β strand) peptides were commercially synthesized (Genscript) at >90% purity level. Peptides were dissolved in water to 400 μM concentration, then diluted into 10 mM sodium-phosphate buffer (pH = 7.5) and 0, 20, or 40 volumetric percent of 2,2,2-trifluoroethanol (TFE) to final concentrations ranging between 15–30 μM. CD spectra were measured at 25 C using a Jasco J-815 Circular Dichroism Spectropolarimeter. The CD spectra were recorded between 180–260 nm with a resolution of 0.5 nm for both peptides and blank buffer solutions in 1 mm cuvettes.

## Computational analyses

Pre-processing steps for high-throughput sequencing were implemented as Snakemake workflows[100]. `Python` (v3.7.4) and `R` (v3.6.2) programming languages were used for all analyses unless mentioned otherwise. In the description below, files ending in `.py` refer to `Python` scripts and files ending in `.Rmd` or `.R` refer to `R Markdown` or `R` scripts.

## Barcode to insert assignment

The raw data from insert-barcode linkage sequencing are in `FASTQ` format. If the inserts and barcodes were on paired-end reads instead of single-end reads, the reads were renamed in increasing numerical order starting at 0 to enable easy matching of insert and barcode reads. This was done in `rename_fastq_paired_reads.py`. The oligo pools were used to create a reference FASTA file in `create_reference_for_aligning_library.R`. A bowtie2[101] (v2.4.2) reference was created from the FASTA file using the `bowtie2-build` command with default options. The insert read was aligned to the

`bowtie2` reference using `bowtie2` command with options `-N 1 -L 22-end-to-end` with the `-trim5` and `-trim3` options set to include only the region corresponding to the insert. The alignments were sorted and indexed using `samtools`[102] (v1.11) commands `sort` and `index` with default options. The alignments were filtered to include only reads with simple `CIGAR` strings and a `MAPQ` score greater than 20 in `filter_alignments.R`. The barcodes corresponding to each filtered alignment were parsed and tallied in `count_barcode_insert_pairs.py`. Depending on the sequencing depth, only barcodes that were observed at least 4-10 times were included in the tally. The tallied barcodes were aligned against themselves using `bowtie2-build` with default options and `bowtie2` with options `-L 24 -N 1 -all -norc`. The self-alignment was used to exclude barcodes that are linked to distinct inserts or ones that are linked to the same barcode but are aligned against each other by `bowtie2`. In the latter case, the barcode with the lower count is discarded in `filter_barcodes.py`. The final list of insert-barcode pairs is written as a tab-delimited `.tsv.gz` file for aligning barcodes from genomic DNA and mRNA sequencing below.

### Barcode counting in genomic DNA and mRNA
The raw data from sequencing barcodes in genomic DNA and mRNA is in FASTQ format. The filtered barcodes `.tsv.gz` file from the insert-barcode linkage sequencing is used to create a reference FASTA file in `create_bowtie_reference.R`. A `bowtie2` (v2.4.2) reference was created from the FASTA file using the `bowtie2-build` command with default options. The barcodes were aligned to the `bowtie2` reference using `bowtie2` command with options `-N 1 -L 20-norc` with the `-trim5` and `-trim3` options set to include only the region corresponding to the barcode. The alignments were sorted, indexed, and tallied using the `samtools` commands `sort`, `index`, `idxstats` with default options. GNU `awk` (v4.1.4) was used for miscellaneous processing of tab-delimited data between pre-processing steps. The final list of counts per barcode in each sample of genomic DNA or mRNA is written as a tab-delimited `.tsv.gz` file for calculating mRNA levels below.

### mRNA quantification
All barcode counts corresponding to each insert in each sample were summed. Only inserts with a minimum of 200 reads and 6 barcodes summed across the mRNA and gRNA samples were included; otherwise the data were designated as missing. mRNA levels were calculated as the log2 ratio of the summed mRNA barcode counts to the summed gRNA barcode counts. mRNA levels were median-normalized within each library. For mRNA stability measurements, the summed mRNA counts for each insert at each time point were normalized by the total barcode counts for the spiked-in HeLa cells at the same time point. Then, the spike-in normalized mRNA levels for each insert were further normalized to the time 0 value.

### Linear statistical modeling of mRNA levels
Amino acid scales for isoelectric point pI, bulkiness, and secondary structure propensity were taken from prior studies[32,36,103]. The median-normalized mRNA levels for lysine, arginine, or glutamate dipeptides were modeled as a function of amino acid scales (as indicated in the figures) using the R function `lm` with default parameters. Only fit coefficients significantly different from zero ($P < 0.05$) are reported for each linear model.

### Secondary structure prediction
Secondary structure was predicted solely from the amino acid sequence using the default single sequence model in S4PRED[35] (downloaded from https://github.com/psipred/s4pred on Apr 17, 2021) and the neural network was used without any modification in `predict_secondary_structure.py`.

Ball and stick representations of 4×SVKF and 4×SKVF in Fig. 4a were predicted using the PEP-FOLD3 server[104] with default parameters and the resulting PDB files were visualized using PyMOL (Schrodinger).

### Calculation of secondary structure content from circular dichroism
The raw circular dichroism data (Fig. 4b, left panel) were converted to the two-column spectrum file format as required for SESCA[105] (v095, downloaded from https://www.mpibpc.mpg.de/sesca on Jul 28, 2021). Secondary structure was estimated using the SESCA script `SESCA_deconv.py` using the pre-computed basis set `Map_BB_DS-dTSC3.dat` and options `@err 2 @rep 100`. The output `.txt` file was parsed to extract the α helix, β strand, and random coil content shown in Fig. 4b, right panel.

### Calculation of ribosome transit time
The raw luminescence vs. time data (Fig. 4c, middle panel) were fit to a straight line in the linear regimes (600 s < t < 900 s for 4×SKVF and 4×KVFS, 900 s < t < 1200 s for 4×SVKF and 4×VKFS) using the R function `lm`. 8×VK luminescence vs. time data (Supplementary Fig. 3b, left panel) were fit in the same manner (linear regimes: 750 s < t < 950 s for 8×VK and 600 s < t < 800 s for +2 Frameshift). The `intercept` term from the fit was used as the transit time of ribosomes across the full transcript and its mean and standard error across technical replicates is shown in the Fig. 4c and Supplementary Fig. 3b right panels.

### Statistical analyses
For barcode sequencing, error bars were calculated as the standard deviation of 100 to 1000 bootstrap samples of barcodes across the gRNA and mRNA samples. The standard deviation was measured for the log2 mRNA levels calculated as described in the *mRNA quantification* section. For all other experiments, the standard error of the mean was calculated using the `std.error` function from the `plotrix` R package. *P*-values for statistically significant differences were calculated using the `t.test` or `wilcox.test` R functions as appropriate for each figure (see figure captions).

### Reporting summary
Further information on research design is available in the Nature Portfolio Reporting Summary linked to this article.

## Data availability
The raw sequencing data generated in this study have been deposited in the Sequence Read Archive under BioProject accession number PRJNA78599. Raw data from circular dichroism and luciferase assays are available at https://github.com/rasilab/burke_2022. Source data are provided with this paper.

## Code availability
Code to reproduce figures in the manuscript starting from raw data is publicly available at https://github.com/rasilab/burke_2022. Requests for biological reagents or clarification can be made by opening an Issue in this repository.

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

## Acknowledgements

We thank members of the Subramaniam lab, the Zid lab, the Basic Sciences Division, and the Computational Biology program at Fred Hutch for discussions and feedback on the manuscript. This research was funded by NIH R35 GM119835, NSF MCB 1846521, and the Sidney Kimmel Scholarship received by ARS. This research was supported by the Genomics Shared Resource of the Fred Hutch/University of Washington Cancer Consortium (P30 CA015704) and Fred Hutch Scientific Computing (NIH grants S10-OD-020069 and S10-OD-028685). The funders had no role in study design, data collection and analysis, decision to publish, or preparation of the manuscript.

## Author contributions

P.C.B. designed research, performed experiments, analyzed data, and wrote the manuscript. H.P. designed research and performed experiments. A.R.S. conceived the project, designed research, analyzed data, wrote the manuscript, supervised the project, and acquired funding.

## Competing interests

The authors declare no competing interests.
