## [Peer Review File · Nature Communications]

A Nascent Peptide Code for Translational Control of mRNA Stability in Human CellsREVIEWER COMMENTS

Reviewer #1 (Remarks to the Author):

In their manuscript entitled "A Nascent Peptide Code for Translation Control of mRNA Stability in Human Cells" Burke et al using a combination of sequencing based screening and measurements of translation elongation and mRNA decay to identify novel motifs that destabilize the mRNA. The authors show that specific combinations of dipeptide repeats lead to substantial differences in mRNA levels. Through different combinations of 4x repeats Serine, Valine, Lysine, Phenylalanine, they can alter the peptide structure from a β -strand to an α -helix. These structural changes lead to ~16 fold changes in mRNA levels, as well as a large decrease in translation elongation speed from the amino acid combinations that formed β -strands. Finally, they explored whether this dipeptide code they identified has an impact on gene expression of endogenous genes. They found that at sites enriched for this dipeptide code there was an increase in ribosome density, indicative of slowing of ribosome elongation, and that these endogenous sequences drove a decrease in mRNA levels when inserted into a reporter.

Overall, this is a very interesting paper that makes novel observations about the importance of amino acid combinations impacting translation elongation and mRNA decay.

Comments:

While this mechanism is described "as a quality control mechanism to test the ability of β -strands to fold into anti-parallel β -sheets and thus avoid aggregation." The question arises of whether these ribosome stalls arise to drive quality control and degrade proteins that are not folding properly or maybe to just slow elongation times to give more time for proper protein folding, or help in the recruitment of chaperones or all of the above. It would be informative to test whether these peptides drive ribosome quality control by using the dual fluorescent reporter from Fig 1A and measuring whether the ratio of RFP to YFP dramatically changes when dipeptide repeats that induce mRNA instability are introduced.

It would also be useful to add to the discussion that slow translation elongation can help recruit protein chaperones (Zhao et al 2021 Genome Biology and Stein et al 2019 Mol Cell), which may be important to properly fold β -strands.

From Figure 3A, mRNAs with dipeptide repeats containing an α -helix are significantly more stable than those grouped as "other". I believe it would be useful for the authors to discuss the possibility that α -helices may stabilize mRNAs or if not postulate why they think that is not the case.

It's unclear if median normalization is the best way to normalize data from Pools 2 and 4 as they have smaller and selected Pools of peptide motifs. For example in Pool 2 this seems to skew the data towards further mRNA destabilization relative to the same data produced in Pool 1. RH, VK, SF all seem to have larger destabilizing effects than in the original library. Would it be better to normalize against a previous value, such as the RH value in Pool 1? It wouldn't change the magnitude of differences in any of the experiments, but could make it easier to compare across experiments.

It could be useful to readers if the authors were able to postulate further into why Arg-His, His-Arg give some of the strongest mRNA destabilizing effects when there is no predicted β -strand formation, and other positively charged amino acid stretches have much lower effects – Arg-Arg, Arg-Lys for example?

Reviewer #2 (Remarks to the Author):

The manuscript from Subramaniam and colleagues builds upon the recent and surprising observation that translational elongation rate impact mRNA stability. For the past few years, it has been established that codon content can impact mRNA decay rates through what is termed codon optimality - the discontinuous rate of decoding based on subtle variations in tRNA concentration. In this manuscript, the authors document that polypeptide sequence can also impact mRNA decay rates, presumably by slowing translational elongation, i.e. the nascent polypeptide is "congesting" the ribosome exit channel. The data are of high quality. The documentation of polypeptide sequence on mRNA stability is clear. The documentation of polypeptide sequence influencing ribosome movement is clear. The only thing that is lacking from this manuscript is the mechanism by which mRNA stability is impacted. Are these results through ribosome stalling, thus a NoGo mRNA decay response? Or does polypeptide sequence impact what is termed "Codon Optimality Mediated mRNA Decay" - this being a CNOT3 response? Alternatively it could be both? Or perhaps something else? It seems like the manuscript would be greatly improved by providing some mechanism. With mechanism, this is a very exciting paper.

Reviewer #3 (Remarks to the Author):

Summary:

In this study, the authors seek to examine the determinants of mRNA stability and links to human gene expression, by considering the primary coding sequence of the mRNA. For this, the authors develop a set of painstaking experiments involving an extensive library coding for dipeptide pairs, CRISPR to enable endogenous expression within human cells, and implementing high-throughput sequencing set-up to quantify mRNA levels by comparing this to genomic DNA. The authors quantify mRNA transcripts and relate this to the primary amino acid sequence of the transcript (as translated peptides). They present evidence of the secondary structure propensity for these peptides, and later, evidence of their impact on translational speed and fidelity. The authors conclude that Lys/Arg (with bulky residues) with beta-strand propensities cause ribosome stalling. They propose a generalized mechanism where such nascent peptides cause ribosome stalling which is in turn, is related to mRNA stability. Overall, this study presents noteworthy results.

Discussion points:

Broadly, the robust technology that the authors have developed is exciting because it offers a means to determine mRNA quantity of a set of genes by comparing this to the levels of genomic DNA. It would seem to me, however, that its output reports on mRNA abundance (steady-state levels), rather than specifically mRNA stability (i.e. half-life); throughout the manuscript the authors use the terms "steady-state mRNA levels" and "mRNA stability" interchangeably. It would be helpful if they could perhaps clarify their nomenclature as necessary (This may appear a trivial concern, but has implications for the author's later deductions).

This study then follows on to present some lovely data including illustrating a relationship between mRNA levels and the primary amino acid sequence of the mRNA transcript. However, beyond this, I am perhaps less convinced that the authors have shown a definitive connection between mRNA stability and ribosome stalling (and thus human gene expression):

1/ In Figure 4C, the authors show changes in translational rates for the dipeptides and later rationalize this in terms of translational arrest (Figure 5B) by re-analysing published ribosome profiling data from HEK293T cells. The original data (Han P et al) already show a propensity for Lys/Arg based motifs inducing ribosome collision/disomes and thus result in ribosomal pausing and very often, arrest. It would be helpful if the authors could show some independent (experimental) evidence of ribosome stalling, since it is a significant finding that they propose from their work (e.g. Line 231, in relation to eukaryotic stall motifs). Along similar lines, it would be helpful for the authors to comment on how

translation rates specifically refer to for ribosome stalling, rather than transient pausing/discontinuous translation.

2/ The authors relate the secondary structure propensity of their dipeptide repeats to ribosome stalling, by discussing possible nascent peptide structure formation and the geometry of the exit tunnel. They use this as a means of rationalizing a stalling mechanism and extrapolate this to global gene expression.

It is highly likely that the dipeptide repeats being considered (12-16 aa) interact within the region between the PTC and uL4/uL22 constriction, to induce pausing or stalling (consistent with translational arrest motifs as observed by cryoEM), however the formation of even very simple tertiary structures (e.g. beta hairpins) typically form beyond the (narrow) constriction. Not all stalling motifs rely on a structured nascent chain either as the authors mentioned. Generally, confirming a stalling mechanism for the new peptides would require more experimental evidence of the ribosome-nascent chain complexes themselves.

I have no doubt that there is a relationship between ribosome stalling and the regulation of gene expression. These links, however, are largely indirect in the data presented. As mentioned above, discerning between mRNA abundance and mRNA stability becomes relevant: For example, ribosome stalling typically induces quality control mechanisms (e.g. RQC), but how this ultimately impacts on an mRNA's stability (i.e. its half-life) or further upstream, its production via transcriptional activation, is not clear from the data. The authors do show actinomycin D data (Figure 2B), but not enough corroborating data showing ribosome stalling and/or protein expression levels.

Perhaps the authors could also have analysed/discussed their data further in the context of related studies that they have referenced? (synonymous codon usage, mRNA structure, GC content etc – mentioned in Line 227 onwards)?

3/ Finally, the authors suggest that the presence of nascent peptides with bulky and extended beta-strand propensities are likely to induce stalling, as a means by which the ribosome avoids aggregation. This scenario is likely to have some nascent peptide length dependence to it, and other considerations too. At least from the current data presented in this study, it is not very clear how (or when) the ribosome might discern a 12-16aa aggregation-prone peptide on its primary sequence alone.

Fundamentally, this is an intriguing study and has the potential to provide some incredible conceptual leaps. I am therefore wholly supportive of the author's intentions. However, some of the experimental evidence feels a little premature to justify the study's major findings.

Response to Reviewers

Summary of changes

We thank all three reviewers for their constructive feedback on our manuscript. We have now performed extensive experiments, analyses, and rewriting of our manuscript to address all their concerns. We believe that these changes significantly improve the rigor of our conclusions and the clarity of our discussion. We highlight below key experiments, analyses, and re-writing in the revised manuscript, which is followed by a detailed point-by-point response.

- 1) To establish protein-level effects for the dipeptide inserts identified to reduce mRNA levels in our study, we performed fluorescence activated cell sorting combined with deep sequencing (FACS-seq) on the HEK293T cell pool expressing our 8x dicodon library (Fig. 1A). This experiment shows that inserts that reduce mRNA levels also cause premature abortive termination (Fig. 2C-D).
- 2) To show that motifs triggering mRNA instability and premature termination also slow ribosome translation, we performed kinetic measurements on reporters encoding a destabilizing dipeptide repeat. These experiments show that the destabilizing dipeptide repeat slows translation elongation relative to a frame-shifted control repeat (Fig. S3).
- 3) To identify the location of ribosomes at the dipeptide peptides identified as reducing mRNA levels, we performed ribosome profiling on mRNAs with synthetic β strand motifs translated using *in vitro* rabbit reticulocyte lysates. Preliminary results from this assay show that the destabilizing β strand motif identified in our study specifically slows down ribosomes when the motif is just fully translated. These results further support our model of ribosome slowdown due to interactions between the nascent peptide and the ribosome exit tunnel.
- 4) To determine the effects of synonymous codon usage, codon optimality, GC and GC3 content in our assay, we systematically correlated related metrics from previous studies with the mRNA effects measured in our massively parallel assay. These analyses show small yet significant correlations with several of these previously determined metrics (Fig. S1). We provide a thorough discussion of these results in the revised manuscript (lines 80–84, lines 209–227).
- 5) As suggested by reviewers, we have extensively updated our Discussion section to address the potential mechanisms by which charged and bulky β strand nascent peptides destabilize mRNA, how these nascent peptide motifs can slow down ribosomes, and why some non β -stranded motifs such as arginine-histidine repeats also reduce mRNA abundance.

Reviewer 1

Summary

In their manuscript entitled “A Nascent Peptide Code for Translation Control of mRNA Stability in Human Cells” Burke et al. using a combination of sequencing based screening and measurements of translation elongation and mRNA decay to identify novel motifs that destabilize the mRNA. The authors show that specific combinations of dipeptide repeats lead to substantial differences in mRNA levels. Through different combinations of 4x repeats Serine, Valine, Lysine, Phenylalanine, they can alter the peptide structure from a β -strand to an α -helix. These structural changes lead to 16 fold changes in mRNA levels, as well as a large decrease in translation elongation speed from the amino acid combinations that formed β -strands. Finally, they explored whether this dipeptide code they identified has an impact on gene expression of endogenous genes. They found that at sites enriched for this dipeptide code there was an increase in ribosome density, indicative of slowing of ribosome elongation, and that these endogenous sequences drove a decrease in mRNA levels when inserted into a reporter.

Overall, this is a very interesting paper that makes novel observations about the importance of amino acid combinations impacting translation elongation and mRNA decay.

We thank the reviewer for recognizing the interesting and novel implications of our work.

Comments

1. While this mechanism is described “as a quality control mechanism to test the ability of β -strands to fold into antiparallel β -sheets and thus avoid aggregation.” The question arises of whether these ribosome stalls arise to drive quality control and degrade proteins that are not folding properly or maybe to just slow elongation times to give more time for proper protein folding, or help in the recruitment of chaperones or all of the above. It would be informative to test whether these peptides drive ribosome quality control by using the dual fluorescent reporter from Fig 1A and measuring whether the ratio of RFP to YFP dramatically changes when dipeptide repeats that induce mRNA instability are introduced.

As suggested by the reviewer, we used our dual fluorescence reporter system to perform FACS-seq (Noderer et al., 2014) on the library of 4096 8x dicodon inserts from Fig. 1A. Dual fluorescent reporters with 2A-linked stall sequences are widely used throughout the translational quality control literature, with abortive premature termination signal used to identify effects on protein expression (Han et al., 2020; Juskiewicz and Hegde, 2017; Sinha et al., 2020; Sundaramoorthy et al., 2017). Reasoning that a low YFP to RFP ratio will indicate abortive translation termination, we used deep sequencing to identify inserts that were enriched in a FACS gate corresponding to low YFP / RFP ratio (Fig. 2C). As a positive control for this assay, we found that all stop codon-containing insert sequences were enriched in this gate since ribosomes will terminate before translation YFP in these constructs (Fig. 2D, reproduced as Fig. RR1

below). Similarly, we found that dipeptide inserts that reduce mRNA levels 2-fold or more below the median in Fig. 1C were also highly enriched in this premature termination bin relative to all other inserts (Fig. 2D, reproduced as Fig. RR1 below). These results show that the dipeptides identified as reducing mRNA levels and mRNA stability in our assay drive abortive premature termination.

Figure RR1 (Fig. 2D in manuscript): Enrichment of dipeptide inserts in the low-YFP gate is shown for three subgroups; inserts encoding in-frame stop codons, dipeptide repeats with mRNA level < -1 (\log_2 , a.u.) in Fig. 1C, and all other inserts. Enrichment values for the RH8, VK8, and SF8 dipeptides are highlighted in orange (RH8, VK8, SF8; left to right). ***: $P < 0.001$ (two-sided Mann-Whitney U test) for differences between subgroups.

2. It would also be useful to add to the discussion that slow translation elongation can help recruit protein chaperones (Zhao et al 2021 Genome Biology and Stein et al 2019 Mol Cell), which may be important to properly fold β -strands.

We have added this point and the related references to our Discussion section (line 305):

Slow translation elongation without mRNA decay can also help recruit protein chaperones, which may be important to properly fold β strands (Stein et al., 2019; Zhao et al., 2021).

3. From Figure 3A, mRNAs with dipeptide repeats containing an α helix are significantly more stable than those grouped as “other”. I believe it would be useful for the authors to discuss the possibility that α -helices may stabilize mRNAs or if not postulate why they think that is not the case.

We believe that β strand formation is destabilizing, rather than α helix formation being stabilizing for the following reasons (discussed in lines 256–261):

First, Fig. 3B shows only aggregates across groups of dipeptides, and such aggregates can result in spurious correlations (Simpson’s paradox). In this specific case, we believe that α helix-forming dipeptides, as a group, show higher mRNA levels due to under-representation of the mRNA-destabilizing arginine and lysine residues in the predicted α helices in our library (Figure S2A, reproduced as Figure RR2 below). This is why we performed the more stringent test of looking at the α helical and β stranded propensities

of individual destabilizing dipeptides in Fig. 3C. This stronger test clearly shows a significant correlation of mRNA destabilization with β -stranded propensity but not with α -helical propensity. We also note the direct relationship between mRNA instability and dipeptide β strand-forming propensity observed in our secondary structure perturbation experiments (Fig. 4D).

Figure RR2 (Fig. S2A in manuscript): Number of dipeptide-encoding reporters from Fig. 1C that contain either arginine or lysine amino acids, partitioned by predicted protein secondary structure as in Fig. 3B.

Second, there are no known mechanisms by which α helix formation stabilizes mRNAs. On the contrary, nascent α helices have been previously postulated to stall ribosomes due to interactions with the ribosome exit tunnel (Bhushan et al., 2010).

Third, we note that a substantial number of predicted unstructured motifs also reduce mRNA level (Fig. 3B). As we note in the Discussion, some unstructured motifs may actually form β strands within the confines of the ribosome exit tunnel that is not captured by the prediction model for isolated peptides. As we also find in our own data, some unstructured motifs can destabilize mRNA without β strand formation (see also point 5, below).

4. It's unclear if median normalization is the best way to normalize data from Pools 2 and 4 as they have smaller and selected Pools of peptide motifs. For example in Pool 2 this seems to skew the data towards further mRNA destabilization relative to the same data produced in Pool 1. RH, VK, SF all seem to have larger destabilizing effects than in the original library. Would it be better to normalize against a previous value, such as the RH value in Pool 1? It wouldn't change the magnitude of differences in any of the experiments, but could make it easier to compare across experiments.

Pool 2 included 1500 motif inserts picked for an unrelated experiment (this was done to economize oligo synthesis). In our initial Pool 2 analysis, we median-normalized to this entire pool, however the majority of these unrelated inserts had no or positive effects, resulting in a more pronounced skew towards lower mRNA levels for our tested destabilizing inserts picked from Pool 1. This was an oversight on our part,

and was not reflected in our description of the methods – we thank the reviewer for pointing out this discrepancy. We have updated our Pool 2 analysis to include only the dipeptide motif inserts described in this work. While the effect magnitudes and trends remain the same, this corrected analysis brings the Pool 2 median-normalized mRNA level values more in line with those measured in Pool 1. However, the absolute mRNA level values for each pool are still imprecise, particularly at lower insert read counts. For this reason, we chose not to re-normalize our pools to the RH8 insert, as the mRNA barcode counts for this insert are low and prone to fluctuation. Thus, we feel that median normalization within pools is more robust, as the median isn't affected by outliers. The remaining differences in mRNA level values for inserts between experiments are likely due to slight differences in growth, cell confluency, RNA harvest conditions, and sequencing depth.

5. It could be useful to readers if the authors were able to postulate further into why Arg-His, His-Arg give some of the strongest mRNA destabilizing effects when there is no predicted β strand formation, and other positively charged amino acid stretches have much lower effects – Arg-Arg, Arg-Lys for example?

As suggested by the reviewer, we have updated our Discussion section to address how the Arg-His repeat might cause such strong effects on mRNA level without β strand formation (lines 261–275):

For example, the arginine-histidine dipeptide repeat destabilizes mRNA and causes premature termination similar to the β stranded Val-Lys and Ser-Phe inserts (Fig. 2A-D). Unlike these latter inserts, Arg-His effects require a longer insert length and strict dipeptide periodicity (Fig. 2E-F), and occur with no predicted β strand formation (Fig. 3A). The 8 \times Arg-His repeats are reminiscent of dipeptide repeat expansions in the human C9ORF72 gene, which cause neurological disease in humans (DeJesus-Hernandez et al., 2011; Mizielinska et al., 2014; Renton et al., 2011). Alternate initiation in the C9ORF72 ORF results in translation of extended Arg-Gly and Arg-Pro repeats, which stall ribosomes and cause premature termination in a length dependent manner, with 20 \times dipeptide repeats being the minimal length required to stall ribosomes (Kriachkov et al., 2022; Loveland et al., 2022). Unsurprisingly, we do not observe marked effects from 8 \times Arg-Gly or Arg-Pro in our assay, as 10 \times repeats of these dipeptides do not cause premature termination (Kriachkov et al., 2022). However, to our knowledge, Arg-His dipeptide repeats have never been tested in this manner prior to our work. It may be that Arg-His repeats impact translation through a similar mechanism as Arg-Gly and Arg-Pro repeats, but with a more acute effect on ribosome elongation that requires fewer repeats to trigger. Notably, recent studies have shown that Arg-rich peptides without Gly or Pro dipeptide periodicity (for example 12 \times Arg) do not stall ribosomes (Kanekura et al., 2018; Kriachkov et al., 2022). This agrees with our finding that positively charged dipeptide repeats composed of 8 \times RR and

8xRK have little effect on mRNA levels (Fig. 1C).

Reviewer 2

Summary

The manuscript from Subramaniam and colleagues builds upon the recent and surprising observation that translational elongation rate impacts mRNA stability. For the past few years, it has been established that codon content can impact mRNA decay rates through what is termed codon optimality - the discontinuous rate of decoding based on subtle variations in tRNA concentration. In this manuscript, the authors document that polypeptide sequence can also impact mRNA decay rates, presumably by slowing translational elongation, i.e. the nascent polypeptide is “congesting” the ribosome exit channel.

Comments

1. The data are of high quality. The documentation of polypeptide sequence on mRNA stability is clear. The documentation of polypeptide sequence influencing ribosome movement is clear.

We thank the reviewer for highlighting the quality and clarity of our work.

2. The only thing that is lacking from this manuscript is the mechanism by which mRNA stability is impacted. Are these results through ribosome stalling, thus a NoGo mRNA decay response? Or does polypeptide sequence impact what is termed “Codon Optimality Mediated mRNA Decay” - this being a CNOT3 response? Alternatively it could be both? Or perhaps something else? It seems like the manuscript would be greatly improved by providing some mechanism. With mechanism, this is a very exciting paper.

We performed the following experiments to address the mechanism by which dipeptide repeats impact mRNA stability:

We have now performed FACS-seq to detect premature abortive translation termination (see response to Reviewer#1, point 2). These experiments indicate that ribosomes prematurely terminate when they translate these destabilizing dipeptide insert sequences (Fig. RR1).

To confirm that the destabilizing inserts from our 8x library screen slow ribosome elongation, we tested the translation kinetics of the VK8 insert characterized in Fig. 2, using the *in vitro* RRL reporter assay from Fig. 4C. The VK8 insert slows ribosome elongation time relative to its frameshifted control to a similar degree as the β -strand inserts in Fig. 4C (~100s). These data are added to the manuscript as Fig. S3B, and reproduced as Figure RR3 below.

Figure RR3 (Fig. S3B in manuscript): *In vitro* measurement of ribosome transit time on mRNAs encoding the 8xVal-Lys dipeptide or its +2 frameshifted control, followed by Nanoluciferase. Luminescence is measured as a function of time after addition of *in vitro* transcribed mRNAs to rabbit reticulocyte lysate (RRL) at $t=0s$ (left panel). Standard error of measurement across three technical replicates is shown as a shaded area on either side of the mean. Ribosome transit times (right panel) are estimated by measuring the X-intercept of the linear portion of the raw luminescence signal in the left panel.

Since the destabilizing dipeptide inserts we identify cause both ribosome slowdown and premature termination, we sought to test whether they trigger ribosome-associated quality control (RQC), which could result in mRNA instability through a corresponding mammalian NoGo decay-like pathway Weber et al. (2020). We discuss this potential mechanism in lines 276–285 of our revised manuscript. To confirm whether RQC factors play a role, we made extensive attempts over past several months to generate CRISPR knockout cells for ZNF598, a key effector for the RQC pathway (Juszkiewicz and Hegde, 2017; Sundaramoorthy et al., 2017). However, our efforts to generate a robust knockout cell line were unsuccessful, with poor antibodies to ZNF598 limiting our ability to detect the presence or absence of this low abundance protein.

It is possible that some dipeptides in our assay cause ribosome slowdown and premature termination without the involvement of RQC. In fact, several examples of pathological peptide repeat sequences that cause ribosome slowdown and premature termination without triggering RQC have been identified in the literature (Kriachkov et al., 2022). This includes Arg-Gly and Arg-Pro dipeptides from *C9ORF72*, which cause amyotrophic lateral sclerosis and frontotemporal dementia (DeJesus-Hernandez et al., 2011; Renton et al., 2011), and poly-glutamine repeats translated from CAG nucleotide repeat expansions in *HTT*, which cause Huntington’s disease (Aviner et al., 2022; Yang et al., 2016; Zheng et al., 2017). Interestingly, although the RQC pathway isn’t demonstrated to act directly on these toxic repeats, expression of RQC pathway components is correlated with limited disease severity in both instances (Aviner et al., 2022; Park et al., 2021). We note these points in our Discussion section (lines 285–291).

To test the alternate mechanism of Ccr4-NOT codon optimality-dependent mRNA decay (Gillen et al., 2021), we reanalyzed our 8x library data to plot the relationship between our mRNA abundance and previous measures of codon optimality. However, we do not observe a relationship between the average

effects of codons in our 8x library with previous measures of codon optimality in HEK293T cells (Figure S1E - reproduced below as Figure RR4). We also did not observe correlation with GC3 content, which has also been postulated to impact mRNA stability in human cells in a CNOT3-dependent manner (Absmeier et al., 2022). Finally, since we do clearly observe premature abortive translation termination at our destabilizing repeats (Fig. RR1), we believe that codon optimality-mediated mRNA decay is unlikely to play a role, since this pathway does not cause premature termination in humans or yeast (Buschauer et al., 2020; Gillen et al., 2021; Veltri et al., 2022).

Figure RR4 (Fig. S1E in manuscript): Average codon effects on mRNA levels plotted from high to low, with codons colored by HEK293T codon optimality (Wu et al., 2019).

Reviewer 3

Summary

In this study, the authors seek to examine the determinants of mRNA stability and links to human gene expression, by considering the primary coding sequence of the mRNA. For this, the authors develop a set of painstaking experiments involving an extensive library coding for dipeptide pairs, CRISPR to enable endogenous expression within human cells, and implementing high-throughput sequencing set-up to quantify mRNA levels by comparing this to genomic DNA. The authors quantify mRNA transcripts and relate this to the primary amino acid sequence of the transcript (as translated peptides). They present evidence of the secondary structure propensity for these peptides, and later, evidence of their impact on translational speed and fidelity. The authors conclude that Lys/Arg (with bulky residues) with beta-strand propensities cause ribosome stalling. They propose a generalized mechanism where such nascent peptides cause ribosome stalling which in turn, is related to mRNA stability. Overall, this study presents noteworthy results.

Comments

1. Broadly, the robust technology that the authors have developed is exciting because it offers a means to determine mRNA quantity of a set of genes by comparing this to the levels of genomic DNA. It would seem to me, however, that its output reports on mRNA abundance (steady-state levels), rather than specifically mRNA stability (i.e. half-life); throughout the manuscript the authors use the terms "steady-state mRNA levels" and "mRNA stability" interchangeably. It would be helpful if they could perhaps clarify their

nomenclature as necessary (This may appear a trivial concern, but has implications for the author's later deductions).

As the reviewer notes, our high-throughput assay reports on steady-state mRNA levels rather than mRNA stability, and we agree that this distinction is important for interpreting our results. We have revised the text to refer specifically to "mRNA stability" or "steady-state mRNA levels (or abundance)" as appropriate for each experiment.

This study then follows on to present some lovely data including illustrating a relationship between mRNA levels and the primary amino acid sequence of the mRNA transcript. However, beyond this, I am perhaps less convinced that the authors have shown a definitive connection between mRNA stability and ribosome stalling (and thus human gene expression)

2. In Figure 4C, the authors show changes in translational rates for the dipeptides and later rationalize this in terms of translational arrest (Figure 5B) by re-analysing published ribosome profiling data from HEK293T cells. The original data (Han P et al) already show a propensity for Lys/Arg based motifs inducing ribosome collision/disomes and thus result in ribosomal pausing and very often, arrest. It would be helpful if the authors could show some independent (experimental) evidence of ribosome stalling, since it is a significant finding that they propose from their work (e.g. Line 231, in relation to eukaryotic stall motifs).

The reviewer correctly notes that ribosome density on mRNAs encoding positively charge and bulky residue combinations is not independent of the Lys/Arg-mediated density previously demonstrated by Han et al. 2020, as well as other studies (Han et al., 2020; Janich et al., 2015; Koppers et al., 2019). In fact, Han and colleagues also observed that not just lysine and arginine residues, but a combination of charged residues interspersed with bulky residues such as phenylalanine and isoleucine, were highly enriched in the P site of stalled ribosomes. Since this observation agrees with our experimental data of charged and bulky residue effects, and we cannot readily distinguish between ribosome density caused by Lys/Arg alone versus Lys/Arg in combination with bulky residues, we have removed this repetition and instead refer to the disome profiling results from Han et al. 2020 (lines 177–182):

Nascent peptides that contain positively charged and bulky amino acids and that are predicted to form β strands trigger ribosome slowdown in human cells. This observation agrees with disome profiling results on endogenous mRNAs, where R-X-K motifs (R – Arg, X – any amino acid, K – lysine) are highly enriched at E, P, and A sites respectively of the lead ribosome (Han et al., 2020). Notably, several R-X-K motifs with the highest disome density have interspersed bulky residues such as phenylalanine, isoleucine, and leucine (Han et al., 2020).

To show independent experimental evidence of ribosome stalling as requested by the reviewer, we per-

formed ribosome profiling on the β strand (SVKF) and α helix-forming (SKVF) peptide reporters from Fig. 4C in rabbit reticulocyte lysates, following the protocol developed by Bhatt et al. 2021 (Bhatt et al., 2021). These reporters differ only in primary sequence and secondary structure but have the same amino acid composition, thus controlling for the effect of any individual amino acid. We observed a specific increase in ribosome density for the β -strand reporter immediately downstream of the 4xSVKF β -strand insert reporter, but not for the 4xSKVF α -helix reporter (Figure RR5 below), suggesting that these peptides are just fully translated before slowing ribosomes. However, a caveat is that most of the sequencing reads underlying these results originated from ribosome footprints in the 35-40 nt range, while typical Ribo-seq footprints are in the 27-35 nt range. We believe that this read length discrepancy might arise from incomplete RNase digestion or from a specific feature of performing Ribo-seq in rabbit reticulocyte lysates. Given these caveats, we chose not to include these results in our manuscript.

Figure RR5: (A) Ribosome profiling of mRNAs encoding α helix or β strand-forming nanoluciferase reporters (from Figure 4C), during *In vitro* translation in rabbit reticulocyte lysate. **(B)** Ribosome profiling read counts from a helix (4xSKVF, top) and β strand (4xSVKF, bottom) reporters. Only reads between 35-40 nucleotides in length were analyzed. **(C)** Zoomed in view of the ribosome density in the region highlighted in blue in **B**. This region encompasses the last 9 amino acids of the α helix (top) or β strand (bottom) inserts.

3. Along similar lines, it would be helpful for the authors to comment on how translation rates specifically

refer to ribosome stalling, rather than transient pausing/discontinuous translation.

As our *in vitro* rabbit reticulocyte ribosome transit assay on its own cannot distinguish stalling from elongation slowdown (Fig. 4C), we have edited the text to refer only to ribosome elongation slowdown when discussing these data. Our preliminary Ribo-seq experiment on *in vitro* mRNAs above suggests that this slowdown occurs at a defined location on the mRNA, but without conclusive measurement of the stall duration relative to other known stalls, we cannot definitively conclude that this slowdown is a stall. Thus, we have used 'ribosome slowdown' consistently throughout the manuscript, except when discussing known stalls.

4a. The authors relate the secondary structure propensity of their dipeptide repeats to ribosome stalling, by discussing possible nascent peptide structure formation and the geometry of the exit tunnel. They use this as a means of rationalizing a stalling mechanism and extrapolate this to global gene expression. It is highly likely that the dipeptide repeats being considered (12-16 aa) interact within the region between the PTC and uL4/uL22 constriction, to induce pausing or stalling (consistent with translational arrest motifs as observed by cryoEM), however the formation of even very simple tertiary structures (e.g. beta hairpins) typically form beyond the (narrow) constriction. Not all stalling motifs rely on a structured nascent chain either as the authors mentioned. Generally, confirming a stalling mechanism for the new peptides would require more experimental evidence of the ribosome-nascent chain complexes themselves.

4b. The authors suggest that the presence of nascent peptides with bulky and extended beta-strand propensities are likely to induce stalling, as a means by which the ribosome avoids aggregation. This scenario is likely to have some nascent peptide length dependence to it, and other considerations too. At least from the current data presented in this study, it is not very clear how (or when) the ribosome might discern a 12-16aa aggregation-prone peptide on its primary sequence alone.

Our responses to the above two comments are related, so we will address them both here.

We agree with the reviewer that confirming a nascent peptide stalling mechanism will require more experimental evidence; ideally cryo-electron microscopy and comprehensive mutational scanning studies of the ribosome-nascent chain complexes. Since these studies would be beyond the scope of the current work, we refrain from claiming a specific mechanism for β strand-mediated stalling in our results. We have edited our Discussion to clarify the points raised by the reviewer on potential arrest mechanisms based on known properties of stalling nascent peptides (Ito and Chiba, 2013; Wilson et al., 2016) as follows:

First, we note the requirement for both charged and bulky amino acids in the primary nascent peptide sequence (lines 228–236):

While positive charge in the nascent peptide can slow ribosomes (Charneski and Hurst, 2013; Lu and Deutsch, 2008), our results show that positive charge by itself is insufficient to induce changes in gene expression in human cells. The importance of bulky amino acids for mRNA stability effects is in line with the role of side chain bulk in ribosome-associated quality control in S. cerevisiae (Mizuno et al., 2021). Further, bulky synthetic amino acid analogs in the nascent peptide and small molecules that add bulk to the exit tunnel can both reduce ribosome elongation rate (Li et al., 2019; Lu et al., 2011; Po et al., 2017; Ramu et al., 2011). Ribosome profiling in S. cerevisiae and human cells shows that tripeptide combinations of bulky and positively charged amino acids are enriched at sites of increased ribosome density (Han et al., 2020; Sabi and Tuller, 2017). Bulky and positively charged amino acids also play critical roles in many known ribosome-arresting peptides (Matheisl et al., 2015; Matsuo et al., 2020; Parola and Kobilka, 1994; Reynolds et al., 1996; Shanmuganathan et al., 2019), and several human arrest peptide sequences stall ribosomes specifically in the presence of small molecule metabolites or drugs in the ribosome exit tunnel (Ivanov et al., 2018; Lintner et al., 2017).

Second, ribosome arrest peptides rely on more than just primary sequence. Known arrest peptides form unique contacts with the ribosome exit tunnel proteins and rRNA; both nascent peptide length and secondary structure inside the exit tunnel are critical for these interactions. We note these in our manuscript as follows (lines 236–239, 241–243):

Structural studies of arrest peptides suggest that bulky and positively charged amino acids might slow down ribosomes by altering the geometry of the peptidyl-transferase center (PTC) and/or by steric interactions with the constriction point in the exit tunnel formed by the uL4 and uL22 proteins (Bhushan et al., 2011; Matsuo et al., 2020; Seidelt et al., 2009; Shanmuganathan et al., 2019).

This role of a simple secondary structural motif like β strand is surprising given that cryo-EM studies of stalled ribosome nascent chain complexes reveal a diverse range of extended conformations, turns, and helices that are specific to each arrest peptide (Li et al., 2019; Matheisl et al., 2015; Matsuo et al., 2020; Su et al., 2017; Wilson et al., 2016).

Finally, while tertiary structures such as β hairpins do not form before the uL4/uL22 constriction point, secondary structures such as α helices, isolated β strands, or unstructured conformations can form within the exit tunnel between the PTC and the constriction (Bhushan et al., 2010; Ito and Chiba, 2013; Khrustalev et al., 2018; Lu and Deutsch, 2005). To discuss this point more clearly, we have updated our previously ambiguous text regarding β strand formation as follows (line 301–303):

Ribosome slowdown at extended β strands could serve as a quality control mechanism, testing the ability of long β strands (10 amino acids or greater in length) to eventually fold into antiparallel β sheets outside the ribosome, and thus avoid aggregation.

We agree with the reviewer that it is likely that these 10-16 aa dipeptide repeats interact within the region between the PTC and uL4/uL22 constriction point in a length-dependent manner. The distance from the construction point to the PTC is about 10 amino acids (~ 30 Å) (Lu and Deutsch, 2005; Lu et al., 2011). Thus, bulky and charged β strands that are 10 amino acids or greater in length may sterically interact with the constriction point and, lacking the ability to compact or flex, transduce this interaction signal back to the PTC (such as for the *SDD1* stall in *S. cerevisiae* (Matsuo et al., 2020)). We note these points in our Discussion (lines 247–251):

At the molecular level, β strands in nascent chains could contribute to ribosome slowdown as an allosteric relay that communicates steric interactions between the nascent chain and the distal portions of the ribosome exit tunnel such as the uL4/uL22 constriction to the PTC (Lu et al., 2011; Matsuo et al., 2020; Seidelt et al., 2009; Wilson et al., 2016; Yap and Bernstein, 2009). This possibility is supported by our observation that destabilizing dipeptide repeats are at least 10-12 amino acids long (Fig. 2C), which is consistent with the distance between the uL4/uL22 constriction and the PTC.

5. I have no doubt that there is a relationship between ribosome stalling and the regulation of gene expression. These links, however, are largely indirect in the data presented. As mentioned above, discerning between mRNA abundance and mRNA stability becomes relevant: For example, ribosome stalling typically induces quality control mechanisms (e.g. RQC), but how this ultimately impacts on an mRNA's stability (i.e. its half-life) or further upstream, its production via transcriptional activation, is not clear from the data. The authors do show actinomycin D data (Figure 2B), but not enough corroborating data showing ribosome stalling and/or protein expression levels.

We now provide corroborating data for effects on ribosome stalling and protein levels as follows:

First, to better characterize the mechanistic link between mRNA stability and protein expression, we have now used FACS-seq to demonstrate that destabilizing dipeptides identified in our assay cause abortive premature termination (Reviewer 1, point 1, Fig. RR1).

Second, we performed Ribo-seq on our RRL reporters to identify specific ribosome slowdown signatures at these dipeptide sequences (Reviewer 3, point 2).

Finally, we have discussed the potential role of RQC pathways in our response to Reviewer 2, point 1. In

addition, we note that, while RQC pathway's effect on mRNA stability is well established in yeast (No-Go decay), the exact relationship between stalling and mRNA stability is not fully understood in human cells (D'Orazio et al., 2019; Hickey et al., 2020; Sinha et al., 2020; Weber et al., 2020), and identity of the intermediate factors that affect mammalian mRNA stability in response to ribosome slowdown remains a subject of ongoing studies (D'Orazio and Green, 2021).

6. Perhaps the authors could also have analysed/discussed their data further in the context of related studies that they have referenced? (synonymous codon usage, mRNA structure, GC content etc – mentioned in Line 227 onwards)?

We thank the reviewer for this useful suggestion. We have now added additional plots of our observed codon, dicodon, and GC content level effects to Fig. S1 (reproduced below as Fig. RR6) and to our results section (80-84). We have interpreted our results in the context of these analyses and previous studies in our Discussion (lines 209–227) as follows:

The nascent peptide code for mRNA stability described here is significantly more complex and localized along the mRNA than previously associated sequence features such as codons, amino acids, and GC content (Forrest et al., 2020; Hia et al., 2019; Narula et al., 2019; Wu et al., 2019). We don't observe large effects on mRNA levels due to codon optimality or GC content in our assay (Fig. S1). This is likely because the 48 nucleotide inserts constitute only ~3% of the 1725 nucleotide coding sequence of our library reporters (Fig. 1A), which limits the impact changing these motifs can have on overall reporter composition. Nevertheless, some individual codon and amino acid signatures in our data agree with previous studies (Fig. 1B, S1). For example, bulky amino acids such as Leu, Ile, Val, and Phe are stabilizing on average, though their codon-specific effects vary across previous studies (Forrest et al., 2020; Narula et al., 2019; Wu et al., 2019). The amino acid serine shows prominent codon-specific effects, with AGU and AGC codons reducing mRNA level more than the remaining codons (Forrest et al., 2020; Hia et al., 2019; Narula et al., 2019; Wu et al., 2019). The methionine AUG start codon and the near-cognate start codons (CUG, GUG, UUG) all promote mRNA stability (Forrest et al., 2020; Hia et al., 2019; Narula et al., 2019; Wu et al., 2019), possibly through effects on increased downstream translation (Wu et al., 2020). With the exception of arginine, lysine, and glycine, our amino acid level effects correlate with the amino acid stability coefficient calculated from endogenous transcripts (Fig S1D) (Forrest et al., 2020). While glycine codons generally stabilize endogenous mRNAs in prior studies, all four glycine codons decrease mRNA levels in our assay, suggesting that glycine dipeptides also cause nascent peptide-mediated ribosome slowdown and mRNA instability. Indeed, we find that Gly-Gly dipeptides reduce mRNA

levels (Fig 1C) consistent with previous observations that poly-glycine motifs stall ribosomes (Chyżyńska et al., 2021). In our data, glycine has the largest effects on mRNA levels when in combination with Leu and Phe, suggesting a nascent peptide-mediated destabilization mechanism akin to that of the biochemically similar Ser-Phe dipeptides.

Figure RR6 (Fig. S1A-D in manuscript): **(A)** Heatmap of mean effect of each codon on mRNA levels averaged across both positions of 8x dicodon repeat. Values >0.50 were set to 0.51, and values <-0.5 were set to -0.51 to highlight differences in intermediate values. **(B)** mRNA levels of dicodon repeats as a function of their GC content (left) or the number of GC3s in the dicodon (right). Spearman correlation coefficient ρ and its P-value are shown for GC content; GC3 content had no significant correlation with mRNA levels. **(C)** Average codon effects on mRNA levels as a function of their mean codon stabilization coefficient (Wu et al., 2019). Pearson correlation coefficient ρ and its P-value are shown. **(D)** Average amino acid effects on mRNA levels of as a function of their mean amino acid stabilization coefficient (Forrest et al., 2020). Pearson correlation coefficient ρ and its P-value are shown, calculated from the points in black. Points in red (arginine, lysine, and glycine) are excluded from this Pearson correlation calculation.

7. Fundamentally, this is an intriguing study and has the potential to provide some incredible conceptual leaps. I am therefore wholly supportive of the author's intentions. However, some of the experimental evidence feels a little premature to justify the study's major findings.

We thank the reviewer for their enthusiastic support for our work. As explained in previous responses,

we have performed additional experiments to substantiate some of our main conclusions. We have also extensively revised our manuscript to ensure that our language accurately describes the results and that our claims are consistent with the presented data. We hope that these changes satisfactorily address the reviewer's concerns.

References

- Absmeier, E., Chandrasekaran, V., O'Reilly, F.J., Stowell, J.A., Rappsilber, J., and Passmore, L.A. (2022). Specific recognition and ubiquitination of slow-moving ribosomes by human CCR4-NOT (Molecular Biology).
- Aviner, R., Lee, T.-T., Masto, V.B., Gestaut, D., Li, K.H., Andino, R., and Frydman, J. (2022). Ribotoxic collisions on CAG expansions disrupt proteostasis and stress responses in Huntington's Disease (Neuroscience).
- Bhatt, P.R., Scaiola, A., Loughran, G., Leibundgut, M., Kratzel, A., Meurs, R., Dreos, R., O'Connor, K.M., McMillan, A., Bode, J.W., et al. (2021). Structural basis of ribosomal frameshifting during translation of the SARS-CoV-2 RNA genome. Science.
- Bhushan, S., Gartmann, M., Halic, M., Armache, J.-P., Jarasch, A., Mielke, T., Berninghausen, O., Wilson, D.N., and Beckmann, R. (2010). \$\alpha\$ -Helical nascent polypeptide chains visualized within distinct regions of the ribosomal exit tunnel. Nat Struct Mol Biol 17, 313–317.
- Bhushan, S., Hoffmann, T., Seidelt, B., Frauenfeld, J., Mielke, T., Berninghausen, O., Wilson, D.N., and Beckmann, R. (2011). SecM-Stalled Ribosomes Adopt an Altered Geometry at the Peptidyl Transferase Center. PLOS Biology 9, e1000581.
- Buschauer, R., Matsuo, Y., Sugiyama, T., Chen, Y.-H., Alhusaini, N., Sweet, T., Ikeuchi, K., Cheng, J., Matsuki, Y., Nobuta, R., et al. (2020). The Ccr4-Not complex monitors the translating ribosome for codon optimality. Science 368.
- Charneski, C.A., and Hurst, L.D. (2013). Positively Charged Residues Are the Major Determinants of Ribosomal Velocity. PLoS Biol. 11, e1001508.
- Chyżyńska, K., Labun, K., Jones, C., Grellscheid, S.N., and Valen, E. (2021). Deep conservation of ribosome stall sites across RNA processing genes. NAR Genomics and Bioinformatics 3, lqab038.
- D'Orazio, K.N., and Green, R. (2021). Ribosome states signal RNA quality control. Mol Cell 81, 1372–1383.

D’Orazio, K.N., Wu, C.C.-C., Sinha, N., Loll-Kripplleber, R., Brown, G.W., and Green, R. (2019). The endonuclease Cue2 cleaves mRNAs at stalled ribosomes during No Go Decay (Molecular Biology).

DeJesus-Hernandez, M., Mackenzie, Ian R., Boeve, Bradley F., Boxer, Adam L., Baker, M., Rutherford, Nicola J., Nicholson, Alexandra M., Finch, NiCole A., Flynn, H., Adamson, J., et al. (2011). Expanded GGGGCC Hexanucleotide Repeat in Noncoding Region of C9ORF72 Causes Chromosome 9p-Linked FTD and ALS. *Neuron* 72, 245–256.

Forrest, M.E., Pinkard, O., Martin, S., Sweet, T.J., Hanson, G., and Collier, J. (2020). Codon and amino acid content are associated with mRNA stability in mammalian cells. *PLoS ONE* 15, e0228730.

Gillen, S.L., Giacomelli, C., Hodge, K., Zanivan, S., Bushell, M., and Wilczynska, A. (2021). Differential regulation of mRNA fate by the human Ccr4-Not complex is driven by coding sequence composition and mRNA localization. *Genome Biol* 22, 284.

Han, P., Shichino, Y., Schneider-Poetsch, T., Mito, M., Hashimoto, S., Udagawa, T., Kohno, K., Yoshida, M., Mishima, Y., Inada, T., et al. (2020). Genome-wide Survey of Ribosome Collision. *Cell Reports* 31, 107610.

Hia, F., Yang, S.F., Shichino, Y., Yoshinaga, M., Murakawa, Y., Vandenbon, A., Fukao, A., Fujiwara, T., Landthaler, M., Natsume, T., et al. (2019). Codon bias confers stability to human mRNAs. *EMBO Reports* 0, e48220.

Hickey, K.L., Dickson, K., Cogan, J.Z., Replogle, J.M., Schoof, M., D’Orazio, K.N., Sinha, N.K., Hussmann, J.A., Jost, M., Frost, A., et al. (2020). GIGYF2 and 4EHP Inhibit Translation Initiation of Defective Messenger RNAs to Assist Ribosome-Associated Quality Control. *Molecular Cell* 79, 950–962.e6.

Ito, K., and Chiba, S. (2013). Arrest Peptides: Cis-Acting Modulators of Translation. *Annu. Rev. Biochem.* 82, 171–202.

Ivanov, I.P., Shin, B.-S., Loughran, G., Tzani, I., Young-Baird, S.K., Cao, C., Atkins, J.F., and Dever, T.E. (2018). Polyamine Control of Translation Elongation Regulates Start Site Selection on Antizyme Inhibitor mRNA via Ribosome Queuing. *Molecular Cell* 70, 254–264.

Janich, P., Arpat, A.B., Castelo-Szekely, V., Lopes, M., and Gatfield, D. (2015). Ribosome profiling reveals the rhythmic liver transcriptome and circadian clock regulation by upstream open reading frames. *Genome Res.* 25, 1848–1859.

Juszkiewicz, S., and Hegde, R.S. (2017). Initiation of Quality Control during Poly(A) Translation Requires Site-Specific Ribosome Ubiquitination. *Mol Cell* 65, 743–750.e4.

- Kanekura, K., Harada, Y., Fujimoto, M., Yagi, T., Hayamizu, Y., Nagaoka, K., and Kuroda, M. (2018). Characterization of membrane penetration and cytotoxicity of C9orf72-encoding arginine-rich dipeptides. *Sci Rep* *8*, 12740.
- Khrustalev, V.V., Khrustaleva, T.A., and Poboinev, V.V. (2018). Amino acid content of beta strands and alpha helices depends on their flanking secondary structure elements. *Biosystems* *168*, 45–54.
- Kriachkov, V., McWilliam, H.E.G., Mintern, J.D., Amarasinghe, S.L., Ritchie, M., Furic, L., and Hatters, D.M. (2022). Arginine-rich C9ORF72 ALS Proteins Stall Ribosomes in a Manner Distinct From a Canonical Ribosome-Associated Quality Control Substrate (Biochemistry).
- Kuppers, D.A., Arora, S., Lim, Y., Lim, A.R., Carter, L.M., Corrin, P.D., Plaisier, C.L., Basom, R., Delrow, J.J., Wang, S., et al. (2019). N⁶-methyladenosine mRNA marking promotes selective translation of regulons required for human erythropoiesis. *Nat Commun* *10*, 1–17.
- Li, W., Ward, F.R., McClure, K.F., Chang, S.T.-L., Montabana, E., Liras, S., Dullea, R.G., and Cate, J.H.D. (2019). Structural basis for selective stalling of human ribosome nascent chain complexes by a drug-like molecule. *Nat Struct Mol Biol* *26*, 501–509.
- Lintner, N.G., McClure, K.F., Petersen, D., Londregan, A.T., Piotrowski, D.W., Wei, L., Xiao, J., Bolt, M., Loria, P.M., Maguire, B., et al. (2017). Selective stalling of human translation through small-molecule engagement of the ribosome nascent chain. *PLOS Biology* *15*, e2001882.
- Loveland, A.B., Svidritskiy, E., Susorov, D., Lee, S., Park, A., Zvornicanin, S., Demo, G., Gao, F.-B., and Korostelev, A.A. (2022). Ribosome inhibition by C9ORF72-ALS/FTD-associated poly-PR and poly-GR proteins revealed by cryo-EM. *Nat Commun* *13*, 2776.
- Lu, J., and Deutsch, C. (2005). Folding zones inside the ribosomal exit tunnel. *Nature Structural & Molecular Biology* *12*, 1123–1129.
- Lu, J., and Deutsch, C. (2008). Electrostatics in the Ribosomal Tunnel Modulate Chain Elongation Rates. *Journal of Molecular Biology* *384*, 73–86.
- Lu, J., Hua, Z., Kobertz, W.R., and Deutsch, C. (2011). Nascent Peptide Side Chains Induce Rearrangements in Distinct Locations of the Ribosomal Tunnel. *Journal of Molecular Biology* *411*, 499–510.
- Matheisl, S., Berninghausen, O., Becker, T., and Beckmann, R. (2015). Structure of a human translation termination complex. *Nucleic Acids Res* *43*, 8615–8626.
- Matsuo, Y., Tesina, P., Nakajima, S., Mizuno, M., Endo, A., Buschauer, R., Cheng, J., Shounai, O., Ikeuchi, K., Saeki, Y., et al. (2020). RQT complex dissociates ribosomes collided on endogenous RQC substrate

SDD1. *Nature Structural & Molecular Biology* 1–10.

Mizielinska, S., Grönke, S., Niccoli, T., Ridler, C.E., Clayton, E.L., Devoy, A., Moens, T., Norona, F.E., Woollacott, I.O.C., Pietrzyk, J., et al. (2014). *C9orf72 repeat expansions cause neurodegeneration in Drosophila through arginine-rich proteins*. *Science* 345, 1192–1194.

Mizuno, M., Ebine, S., Shounai, O., Nakajima, S., Tomomatsu, S., Ikeuchi, K., Matsuo, Y., and Inada, T. (2021). *The nascent polypeptide in the 60S subunit determines the Rqc2-dependency of ribosomal quality control*. *Nucleic Acids Research*.

Narula, A., Ellis, J., Taliaferro, J.M., and Rissland, O.S. (2019). *Coding regions affect mRNA stability in human cells*. *RNA* 25, 1751–1764.

Noderer, W.L., Flockhart, R.J., Bhaduri, A., Arce, A.J.D. de, Zhang, J., Khavari, P.A., and Wang, C.L. (2014). *Quantitative analysis of mammalian translation initiation sites by FACS-seq*. *Molecular Systems Biology* 10, 748.

Park, J., Lee, J., Kim, J., Lee, J., Park, H., and Lim, C. (2021). *ZNF598 co-translationally titrates poly(GR) protein implicated in the pathogenesis of C9ORF72 -associated ALS/FTD*. *Nucleic Acids Research* 49, 11294–11311.

Parola, A.L., and Kobilka, B.K. (1994). *The peptide product of a 5' leader cistron in the beta 2 adrenergic receptor mRNA inhibits receptor synthesis*. *Journal of Biological Chemistry* 269, 4497–4505.

Po, P., Delaney, E., Gamper, H., Szantai-Kis, D.M., Speight, L., Tu, L., Kosolapov, A., Petersson, E.J., Hou, Y.-M., and Deutsch, C. (2017). *Effect of Nascent Peptide Steric Bulk on Elongation Kinetics in the Ribosome Exit Tunnel*. *Journal of Molecular Biology* 429, 1873–1888.

Ramu, H., Vázquez-Laslop, N., Klepacki, D., Dai, Q., Piccirilli, J., Micura, R., and Mankin, A.S. (2011). *Nascent Peptide in the Ribosome Exit Tunnel Affects Functional Properties of the A-Site of the Peptidyl Transferase Center*. *Molecular Cell* 41, 321–330.

Renton, Alan E., Majounie, E., Waite, A., Simón-Sánchez, J., Rollinson, S., Gibbs, J. Raphael., Schymick, Jennifer C., Laaksovirta, H., van Swieten, John C., Myllykangas, L., et al. (2011). *A Hexanucleotide Repeat Expansion in C9ORF72 Is the Cause of Chromosome 9p21-Linked ALS-FTD*. *Neuron* 72, 257–268.

Reynolds, K., Zimmer, A.M., and Zimmer, A. (1996). *Regulation of RAR beta 2 mRNA expression: evidence for an inhibitory peptide encoded in the 5'-untranslated region*. *Journal of Cell Biology* 134, 827–835.

Sabi, R., and Tuller, T. (2017). *Computational analysis of nascent peptides that induce ribosome stalling*

and their proteomic distribution in *Saccharomyces cerevisiae*. *RNA* 23, 983–994.

Seidelt, B., Innis, C.A., Wilson, D.N., Gartmann, M., Armache, J.-P., Villa, E., Trabuco, L.G., Becker, T., Mielke, T., Schulten, K., et al. (2009). Structural Insight into Nascent Polypeptide Chain–Mediated Translational Stalling. *Science* 326, 1412–1415.

Shanmuganathan, V., Schiller, N., Magoulopoulou, A., Cheng, J., Braunger, K., Cymer, F., Berninghausen, O., Beatrix, B., Kohno, K., von Heijne, G., et al. (2019). Structural and mutational analysis of the ribosome-arresting human XBP1u. *eLife* 8, e46267.

Sinha, N.K., Ordureau, A., Best, K., Saba, J.A., Zinshteyn, B., Sundaramoorthy, E., Fulzele, A., Garshott, D.M., Denk, T., Thoms, M., et al. (2020). EDF1 coordinates cellular responses to ribosome collisions. *eLife* 9, e58828.

Stein, K.C., Kriel, A., and Frydman, J. (2019). Nascent Polypeptide Domain Topology and Elongation Rate Direct the Cotranslational Hierarchy of Hsp70 and TRiC/CCT. *Molecular Cell* 75, 1117–1130.e5.

Su, T., Cheng, J., Sohmen, D., Hedman, R., Berninghausen, O., von Heijne, G., Wilson, D.N., and Beckmann, R. (2017). The force-sensing peptide VemP employs extreme compaction and secondary structure formation to induce ribosomal stalling. *eLife* 6, e25642.

Sundaramoorthy, E., Leonard, M., Mak, R., Liao, J., Fulzele, A., and Bennett, E.J. (2017). ZNF598 and RACK1 Regulate Mammalian Ribosome-Associated Quality Control Function by Mediating Regulatory 40S Ribosomal Ubiquitylation. *Molecular Cell* 65, 751–760.e4.

Veltri, A.J., D’Orazio, K.N., Lessen, L.N., Loll-Krippléber, R., Brown, G.W., and Green, R. (2022). Distinct elongation stalls during translation are linked with distinct pathways for mRNA degradation. *eLife* 11, e76038.

Weber, R., Chung, M.-Y., Keskeny, C., Zinnall, U., Landthaler, M., Valkov, E., Izaurralde, E., and Igreja, C. (2020). 4EHP and GIGYF1/2 Mediate Translation-Coupled Messenger RNA Decay. *Cell Reports* 33, 108262.

Wilson, D.N., Arenz, S., and Beckmann, R. (2016). Translation regulation via nascent polypeptide-mediated ribosome stalling. *Current Opinion in Structural Biology* 37, 123–133.

Wu, C.C.-C., Peterson, A., Zinshteyn, B., Regot, S., and Green, R. (2020). Ribosome Collisions Trigger General Stress Responses to Regulate Cell Fate. *Cell* 182, 404–416.e14.

Wu, Q., Medina, S.G., Kushawah, G., DeVore, M.L., Castellano, L.A., Hand, J.M., Wright, M., and Bazzini, A.A. (2019). Translation affects mRNA stability in a codon-dependent manner in human cells. *eLife* 8,

e45396.

Yang, J., Hao, X., Cao, X., Liu, B., and Nyström, T. (2016). Spatial sequestration and detoxification of Huntingtin by the ribosome quality control complex. *eLife* 5, e11792.

Yap, M.-N., and Bernstein, H.D. (2009). The Plasticity of a Translation Arrest Motif Yields Insights into Nascent Polypeptide Recognition inside the Ribosome Tunnel. *Molecular Cell* 34, 201–211.

Zhao, T., Chen, Y.-M., Li, Y., Wang, J., Chen, S., Gao, N., and Qian, W. (2021). Disome-seq reveals widespread ribosome collisions that promote cotranslational protein folding. *Genome Biology* 22, 16.

Zheng, J., Yang, J., Choe, Y.-J., Hao, X., Cao, X., Zhao, Q., Zhang, Y., Franssens, V., Hartl, F.U., Nyström, T., et al. (2017). Role of the ribosomal quality control machinery in nucleocytoplasmic translocation of polyQ-expanded huntingtin exon-1. *Biochemical and Biophysical Research Communications* 493, 708–717.

REVIEWERS' COMMENTS

Reviewer #1 (Remarks to the Author):

I am satisfied by the changes made by the authors and recommend publication of this manuscript.

Reviewer #2 (Remarks to the Author):

The authors have satisfied my concerns. This is an exciting paper.

Reviewer #3 (Remarks to the Author):

The authors have addressed my queries very well, and I would also like to commend them in their efforts to elevate this study even further.